# Efficient Online Pruning and Abstraction for Imperfect Information Extensive-Form Games

**Boning Li**
Institute for Interdisciplinary Information Sciences, Tsinghua University
`li-bn22@mails.tsinghua.edu.cn`

**Longbo Huang** *
Institute for Interdisciplinary Information Sciences, Tsinghua University
`longbohuang@tsinghua.edu.cn`

## Abstract

Efficiently computing approximate equilibrium strategies in large Imperfect Information Extensive-Form Games (IIEFGs) poses significant challenges due to the game tree's exponential growth. While pruning and abstraction techniques are essential for complexity reduction, existing methods face two key limitations: (i) Seamless integration of pruning with Counterfactual Regret Minimization (CFR) is nontrivial, and (ii) Pruning and abstraction approaches incur prohibitive computational costs, hindering real-world deployment. We propose Expected-Value Pruning and Abstraction (EVPA), a novel online framework that addresses these challenges through three synergistic components: (i) Expected value estimation using approximate Nash equilibrium strategies to quantify information set utilities, (ii) Minimax pruning before CFR to eliminate a large number of sub-optimal actions permanently, and (iii) Dynamic online information abstraction merging information sets based on their current and future expected values in subgames. Experiments on Heads-up No-Limit Texas Hold'em (HUNL) show EVPA outperforms DeepStack's replication and Slumbot with significant win-rate margins in multiple settings. Remarkably, EVPA requires only 1%-2% of the solving time to reach an approximate Nash equilibrium compared to DeepStack's replication.

## 1 Introduction

Imperfect Information Extensive-Form Games (IIEFGs) provide a robust framework for analyzing sequential games with hidden information and multiple players. This framework is applicable across various domains, such as Poker (Brown & Sandholm, 2019a), Mahjong (Li et al., 2020), and Stratego (Perolat et al., 2022). Counterfactual Regret Minimization (CFR) and its variants (Zinkevich et al., 2007b; Lanctot et al., 2009; Tammelin, 2014; Brown et al., 2019; Brown & Sandholm, 2019b; Xu et al., 2024) stand out as the leading approaches for solving IIEFGs. However, the computational overhead of CFR scales with the size of the game tree, making it challenging to compute approximate equilibrium strategies for large IIEFGs, particularly in games such as Heads-Up No-Limit Texas Hold'em (HUNL), which features a game tree with roughly $10^{165}$ states (Johanson, 2013).

Reducing the size of the game tree is essential for making equilibrium computation feasible (Sandholm, 2010). Pruning techniques (Blair et al., 1996) eliminate sub-optimal branches, speeding up CFR convergence and reducing computational overhead (Brown & Sandholm, 2015a). Similarly, abstraction techniques (Sandholm, 2015) group similar information sets into buckets, significantly shrinking the game tree size. Combining pruning and abstraction methods can substantially reduce the game tree size, making CFR more practical for large IIEFGs (Brown & Sandholm, 2016b; 2018).

Despite their utility, current pruning and information abstraction techniques have limitations. Existing pruning methods often depend on intermediate computed values during CFR iterations (Lanctot et al., 2009; Brown & Sandholm, 2015a; 2017a), leading to dynamic and tentative pruning. This

---

*Corresponding author.

complexity requires tailored adjustments based on the specific CFR variant employed. More importantly, when using generic techniques such as depth-limited solving (Moravčík et al., 2017) or MCCFR (Lanctot et al., 2009), the computational overhead of these intermediate values may even greatly exceed the original overhead of the iteration. Moreover, during early CFR iterations, the game tree size remains unchanged, resulting in no reduction in memory usage.

Information abstraction methods can be broadly categorized into expectation-based abstraction (Gilpin & Sandholm, 2007; Zinkevich et al., 2007a) and potential-aware abstraction (Gilpin et al., 2007; 2008; Ganzfried & Sandholm, 2014). Expectation-based abstraction often neglects the future evolution of information sets, making it less effective (Gilpin & Sandholm, 2008; Johanson et al., 2013). While potential-aware abstraction is more comprehensive, it requires extensive simulation and clustering, leading to significant computational overhead that can extend for months (Sandholm, 2010; Brown et al., 2015), making it impractical for online computation. Furthermore, when utilizing subgame solving techniques (Ganzfried & Sandholm, 2015; Brown & Sandholm, 2017c), previous methods often use the same pre-calculated abstraction across different subgames, which can be sub-optimal.[1] Additionally, many methods focus solely on the strength of information sets, overlooking blocking effects (Sandholm, 2010).[2]

To address these challenges, we propose Expected-Value Pruning and Abstraction (EVPA), a novel online method that integrates expected value estimation into pruning and abstraction processes. The goal is to significantly reduce the solving time required to reach an $\varepsilon$-Nash equilibrium. EVPA consists of three core components: expected value estimation of information sets, expected value-based pruning, and information abstraction for subgames. Figure 1 illustrates how EVPA method operates in a HUNL subgame example.

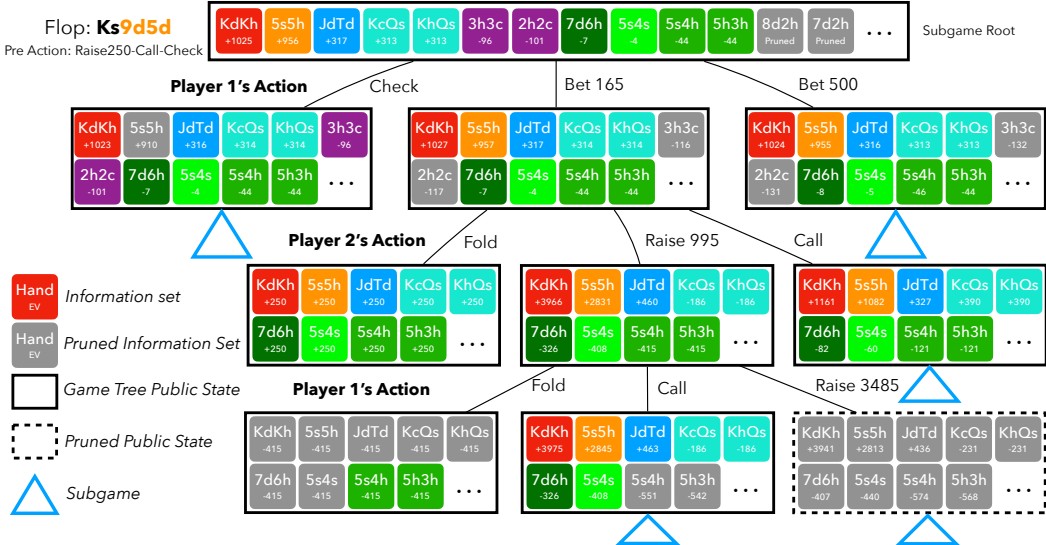

Figure 1: In the HUNL example with 200 big blinds (20,000 chips), the player in the small blind (Player 1) raises to 250 chips, and the player in the big blind (Player 2) calls during the pre-flop stage. The flop reveals Ks9d5d, after which Player 2 checks, making it Player 1's turn to act. The figure illustrates a portion of the current subgame constructed by EVPA. First, EVPA estimates the expected value of each information set within the subgame. It then prunes branches with significantly lower expected values, as indicated by gray blocks representing pruned information sets. Finally, EVPA clusters the remaining information sets based on their current and future expected values. Information sets of the same color in the figure are grouped into the same bucket, where a "bucket" refers to a collection of information sets that share the same strategy within the subgame.

---

[1]For example, in HUNL, holding a bottom pair may be advantageous if both players check to the river, but can be weaker if significant pot increases occur before the river.

[2]For instance, in HUNL, when the public cards on the river are KsTs9d6c5s, holding As3h and holding Ac3h have equal strength, but it is less likely for the opponent to hold the nuts (the strongest hand) with As3h.

The first component, expected value estimation of information sets, generates expected value estimations for each information set in the subgame, based on the approximate Nash equilibrium strategy. EVPA's estimation does not rely on the probability distributions of information sets, allowing it to efficiently sample data and avoid the overhead of probability distribution calculations typically required during CFR iterations (Kroer & Sandholm, 2015; Brown et al., 2018). The innovation of EVPA lies in its ability to effectively harness these expected value estimations specifically for both pruning and information abstraction before the CFR process begins. This capability optimizes the decision-making process, enhancing computational efficiency while maintaining strategic depth.

The second component, expected value-based pruning, employs Minimax pruning (Blair et al., 1996) based on the expected value estimation of information sets. By permanently and correctly eliminating sub-optimal branches before the CFR begins, EVPA enables the CFR to concentrate on the most important branches, leading to a substantial increase in convergence speed. Notably, EVPA's pruning is efficient and does not require real-time computation of intermediate values, unlike previous methods (Brown & Sandholm, 2015a; 2017a; Brown et al., 2017). Additionally, it can be seamlessly integrated with various CFR variants (Lanctot et al., 2009; Brown & Sandholm, 2019b).

The third component, information abstraction for subgames, introduces a novel online algorithm that merges information sets based on both current and future expected values. Compared to previous methods (Gilpin & Sandholm, 2007; Ganzfried & Sandholm, 2014), EVPA's information abstraction considers relative strength, blocking effects, and potential strength of the information sets, while achieving efficient abstraction in under 1 second. EVPA's abstraction is effective in online CFR solving and is scalable to larger IIEFGs such as Omaha (Farha & Reback, 2007). Furthermore, the flexibility in selecting bucket sizes allows EVPA to balance solving time and abstraction granularity.

We evaluate EVPA using Heads-up No-Limit Texas Hold'em (HUNL) poker and compare its performance against DeepStack's replication (Moravčík et al., 2017) and strong open-source AI Slumbot (Jackson, 2013). EVPA shows significant improvements, including reductions in game tree sizes ranging from $42.67\%$ to $79.51\%$ across different abstraction settings and subgames. Notably, EVPA reduces the solving time required to reach an $\varepsilon$-Nash equilibrium to just 1%-2% of the time needed for DeepStack's replication. EVPA also outperformed the DeepStack's replication with win-rates of $930 \pm 23, 202 \pm 31, 82 \pm 60$ mbb/h when the information set is touched $1 \times 10^7$, $1 \times 10^8$ and $1 \times 10^9$ times, respectively. Additionally, EVPA beats Slumbot with win-rates of $10 \pm 26, 96 \pm 43, 100 \pm 68$ mbb/h under 0.02, 0.2 and 2 seconds of the solving time limits, respectively.

The main contributions of our work are as follows:

- *Novel Expected Valued-Based Pruning.* We introduce a highly effective pruning technique that permanently and correctly eliminates sub-optimal actions from the game tree before the CFR process begins. This approach achieves up to 98.6% reduction in exploitability in HUNL subgames, enabling us to avoid traversing unnecessary paths. Notably, this pruning method does not require complex computations and exhibits excellent scalability.

- *Advanced Information Abstraction for Subgames.* Our information abstraction method provides efficient abstractions that consider both current and future expected values, enabling tailored online abstractions for different subgames with minimal computational overhead. This abstraction method accelerates convergence significantly in HUNL subgames, and has the potential to be applied to more complex games than HUNL.

- *Super Performance on HUNL.* EVPA effectively integrates the core techniques from previous superhuman performance HUNL AIs, such as Libratus (Brown & Sandholm, 2018) and DeepStack. Experiments with limited solving time on HUNL demonstrate that EVPA surpasses both DeepStack's replication and strong bot Slumbot with significant win-rate margins, indicating that EVPA can achieve super performance with minimal solving time.

## 2 RELATED WORK

**Value Estimation in IIEFGs.** Value estimation is predominantly used to substitute leaf node values in depth-limited solving. DeepStack (Moravčík et al., 2017) achieved superhuman performance in HUNL through probability distribution-based value estimation during CFR iterations. Subsequent works like Supremus (Zarick et al., 2020) and ReBeL (Brown et al., 2020), further validated the

reliability of this approach. Another method for estimating leaf node values involves selecting the highest value from multiple strategies (Brown et al., 2018). Beyond depth-limited solving, value estimation has applications in various IIEFG techniques, including Deep CFR (Brown et al., 2019), action abstraction (Li et al., 2024) and variance reduction (Burch et al., 2018).

**Pruning with CFR.** In CFR, the value and reach probability of an information set change with each iteration. This dynamic nature means that most previous pruning methods rely on temporary pruning using intermediate computed values. For instance, regret-based pruning (Brown & Sandholm, 2015a) avoids traversing a path if either player takes actions leading to that path with zero probability. Furthermore, best-response pruning (Brown & Sandholm, 2017a) allows for the temporary pruning of poorly performing actions. Additionally, partial pruning (Lanctot et al., 2009) and dynamic thresholding pruning (Brown et al., 2017) enable the pruning of actions with low probability. A more detailed description of pruning methods in IIEFGs can be found in Appendix G.

**Information Abstraction in IIEFGs.** Information abstraction can be categorized into expectation-based abstraction and potential-aware abstraction (Gilpin & Sandholm, 2008). Expectation-based abstraction methods (Gilpin & Sandholm, 2007; Zinkevich et al., 2007a) classify information sets based on their current strength of expectation, while potential-aware abstraction methods (Gilpin et al., 2007; 2008) consider the performance of information sets across different future scenarios. The imperfect-recall technique (Waugh et al., 2009; Ganzfried & Sandholm, 2014) allows players to intentionally forget certain information, significantly reducing the size of the game tree. A more detailed description of the abstraction can be found in Appendix H.

## 3 BACKGROUND AND NOTATION

In an Imperfect Information Extensive-Form Game (IIEFG), there is a finite set of players $\mathcal{N} = \{1, \cdots, N\}$. A state, also known as history $h$, represents the sequence of all historical actions taken from the initial state $\emptyset$. Performing an action $a$ in a non-terminating state $h$ transitions to a new state $h'$, denoted as $h \cdot a = h'$, where $h$ is the parent of $h'$. If state $h'$ can be reached from $h$ by performing a sequence of actions, then $h$ is an ancestor of $h'$. We use the notation $h \sqsubseteq h'$ to mean $h$ is an ancestor of $h'$, and $h \sqsubset h'$ to mean $h$ is a strict ancestor of $h'$. A terminal state $z$ is one where no further actions are available, and $u_p(z)$ represents the payoff for player $p$ at terminal state $z$. The acting player at a non-terminal state $h$ is denoted by $\mathcal{P}(h) \in \mathcal{N} \cup \{c\}$, where $c$ represents the "chance player," indicating events beyond the control of players in $\mathcal{N}$.

For each player $p \in \mathcal{N}$, imperfect information is represented by an information set $I_p$. All states $h, h' \in I_p$ are indistinguishable to $p$. If $p$ is the acting player, the information set $I_p$ can be simply denoted as $I$. The set of information sets is denoted by $\mathcal{I}$. A strategy $\sigma(I)$ is a probability distribution over the available actions within an information set $I$. The probability of choosing action $a$ at information set $I$ is denoted as $\sigma(I, a)$. The strategy for player $p$ in all information sets where they act is denoted as $\sigma_p$, while the strategies for all other players are denoted as $\sigma_{-p}$. A strategy profile $\sigma = (\sigma_p)_{p \in \mathcal{N}}$ is a collection of strategies, one for each player. The expected value (EV) of an information set $I_p$ under a strategy profile $\sigma$ is denoted $EV_\sigma(I_p)$.

The best response for player $p$ to $\sigma_{-p}$ is a strategy that maximizes player $p$'s payoff. Mathematically, $BR(\sigma_{-p}) = \arg\max_{\sigma'_p} u_p(\sigma'_p, \sigma_{-p})$. A Nash equilibrium $\sigma^*$ is a strategy profile where each player's strategy is a best response to the strategies of the other players. That is, $\forall p, u_p(\sigma^*, \sigma^*_{-p}) = \max_{\sigma'_p} u_p(\sigma'_p, \sigma^*_{-p})$. The exploitability $e(\sigma_p)$ of a strategy $\sigma_p$ in a two-player zero-sum game measures how much worse the strategy performs compared to a best response against a Nash equilibrium strategy. Formally, $e(\sigma_p) = u_p(\sigma^*_p, BR(\sigma^*_p)) - u_p(\sigma_p, BR(\sigma_p))$. A strategy profile $\sigma$ is an $\varepsilon$-Nash equilibrium if no player has an exploitability greater than $\varepsilon$ under $\sigma$.

A subgame is a continuous portion of a game tree. Formally, a subgame $S$ is a set of states such that $\forall h \in S$, if $h \in I_p$ and $h' \in I_p$ then $h' \in S$, and $\forall x, z \in S$, if $x \sqsubset y \sqsubset z$ then $y \in S$. A public state (or node) $s$ contains the information known to all players. The unique public state corresponding to a state $h$ and an information set $I_p$ is denoted as $s(h)$ and $s(I_p)$, respectively. If a state $h \in S$ has no descendants within $S$, it is called a leaf state, and the information sets and nodes containing $h$ are called leaf information sets and leaf nodes. Conversely, if $h \in S$ has no ancestors within $S$, it is called a root state, and the information sets and nodes containing $h$ are called root information sets and root nodes. The root node of $S$ is denoted as $S_r$.

# 4 EVPA ALGORITHM

In this section, we present the Expected-Value Pruning and Abstraction (EVPA) algorithm. The EVPA algorithm is composed of three core components: (i) expected value estimation of information sets (Section 4.1), which calculates the expected value of each information set to serve as a foundation for subsequent operations; (ii) expected value-based pruning (Section 4.2), where suboptimal branches are permanently removed from the game tree before CFR using the estimated expected values; and (iii) information abstraction for subgames (Section 4.3), which groups similar information sets based on their current and future expected values in subgames to reduce complexity. Figure 2 visually depicts the operation of these three components in a depth-limited subgame.

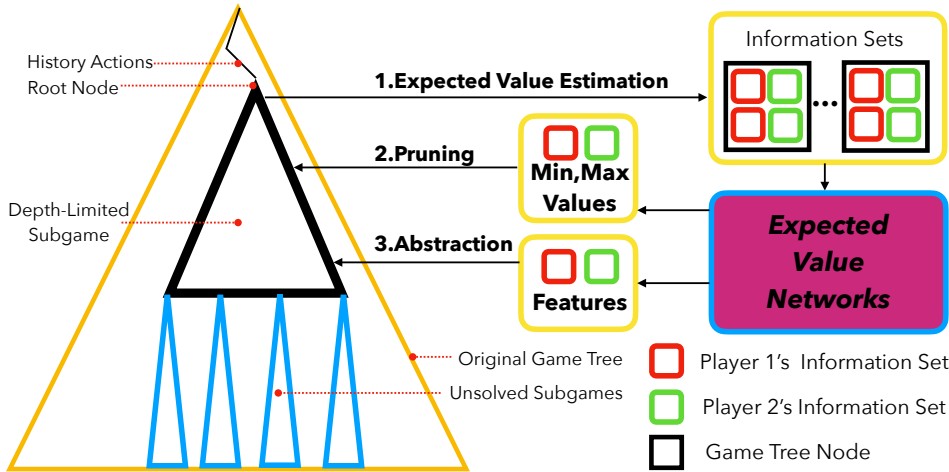

Figure 2: Framework of EVPA. We begin by estimating the maximum, minimum, and average expected values of each information set in the subgame using expected value networks. Next, we prune the game tree based on the maximum and minimum values. Finally, we utilize the average values as features for information abstraction.

## 4.1 EXPECTED VALUE ESTIMATION OF INFORMATION SETS

In this subsection, we detail the expected value estimation process of information sets within the EVPA algorithm. We begin by using a framework similar to DeepStack's (Moravčík et al., 2017) for computing depth-limited subgame equilibrium strategies (further implementation details are available in Appendix D). This framework is trained using a large dataset of game states and actions from a wide variety of IIEFG scenarios. Once the training process is completed, we leverage this trained framework to estimate the expected values of approximate Nash equilibrium strategies for various information sets in IIEFGs.

Our method involves sampling diverse subgames and calculating approximate Nash equilibrium expected values for information sets within these subgames. The sampling process is outlined in Algorithm 1. We start by generating a depth-limited subgame $S$ from a random initial public state. To ensure comprehensive action coverage, we utilize randomized action abstractions during subgame construction. We then compute the Nash equilibrium strategy for this subgame, calculate the expected value for each information set, and incorporate these values into our training data. The process continues by sampling a leaf node $s$ of the subgame $S$. If $s$ is not a terminal node, we then solve the subgame rooted at $s$. We emphasize that, unlike DeepStack, the information set features in EVPA contain only public information and player's private hand information, and do not include range information of both players.

Once we have collected sufficient data through the sampling process outlined in Algorithm 1, we proceed to train neural networks for the purpose of expected value estimation. We train $M$ independent neural networks (with $M = 10$) using the Huber Loss (Huber, 1964) as our loss function. Each neural network receives a feature vector describing the information set, and outputs a scalar representing the expected value (relative to the pot size) of that information set.

**Algorithm 1:** Algorithm for Sampling of Nash Equilibrium Expected Value of Information Sets.

$Data \leftarrow \emptyset, s \leftarrow$ RandomInitPublicState()// `Randomly initialization`
**while** $\mathcal{P}(s) \neq \emptyset$ **do**
    **while** $\mathcal{P}(s) = c$ **do**
        $s \leftarrow$ TakeChance($s$)// `Process a random chance event`
    $S \leftarrow$ DepthLimitedSubgame($s$)// `Build a depth-limited subgame with`
        `randomized action abstractions`
    $\sigma^* \leftarrow$ StrategySolving($S$) // `Compute Nash equilibrium strategies`
    **for** $I_p \in S$ **do**
        $EV_{\sigma^*}(I_p) \leftarrow$ CalculateExpectedValue($S, \sigma^*, I_p$)
        Add $\{I_p, EV_{\sigma^*}(I_p)\}$ to $Data$
    $s \leftarrow$ SampleLeaf($S, \sigma^*$) // `Randomly sampling a leaf`
**Output:** $Data$

The network architecture consists of 6 layers of Multi-Layer Perceptrons (MLPs), each with $1,536$ hidden units and ReLU activation functions (Glorot et al., 2011). Each network is trained on at least 10 billion samples using the Adam optimizer (Kingma & Ba, 2015). Each sample provides values for all information sets within subgame $S$, resulting in a sampling size that is millions of times larger than the one used by DeepStack within the same sampling time. Training the DeepStack replication requires 70 days, with sampling time being the primary overhead. In contrast, training the expected value network in EVPA takes only 4 days, with 1 day of sampling time.

Upon completing the training, we obtain $M$ expected value estimation networks. For each information set $I_p$, the expected value estimation from the $i$-th network is denoted as $EV_i(I_p)$. We calculate the average expected value estimation as $\overline{EV}(I_p) = \frac{1}{M} \sum_{i=1}^{M} EV_i(I_p)$. Additionally, we determine the maximum and minimum expected value estimations across the $M$ networks as $EV_{max}(I_p) = \max\{EV_1(I_p), \ldots, EV_M(I_p)\}$ and $EV_{min}(I_p) = \min\{EV_1(I_p), \ldots, EV_M(I_p)\}$, respectively. The computed average, maximum, and minimum expected values will serve as crucial inputs for the pruning and abstraction steps, allowing us to make informed decisions on which parts of the game tree to prune or abstract. The true innovation of EVPA lies in its effective use of these expected value insights, enabling smarter decision-making and more efficient exploration of the game tree. This ultimately enhances the overall performance of the algorithm.

## 4.2 EXPECTED VALUE-BASED PRUNING

In this subsection, we introduce the Expected Value-based Pruning component of the EVPA algorithm, which is a crucial part of our approach to reduce the computational complexity of IIEFGs. This pruning technique uses the expected value estimates from our previously trained networks to make informed decisions about which parts of the game tree can be safely pruned.

The EVPA pruning method introduces a novel approach inspired by Minimax pruning (Blair et al., 1996), based on the principle that an optimal player will not select an action for an information set if another action has a higher expected value. For instance, in HUNL, discarding a pair of Aces pre-flop is clearly sub-optimal compared to calling or raising, which yield higher expected values. Formally, for an information set $I$, an action $a$, and a Nash equilibrium strategy $\sigma^*$, if $EV_{\sigma^*}(I \cdot a) < EV_{\sigma^*}(I)$, then $\sigma^*(I, a) = 0$. Furthermore, if there exists another action $a'$ such that $EV_{\sigma^*}(I \cdot a) < EV_{\sigma^*}(I \cdot a')$, then $\sigma^*(I, a) = 0$.

However, accurately calculating exactly expected value under Nash equilibrium can be computationally prohibitive. To address this challenge, the EVPA method leverages estimates from $M$ expected value estimation networks to perform pruning efficiently before applying the CFR algorithm. This integration enhances the effectiveness of pruning and significantly reduces computational overhead.

The details of the pruning algorithm are outlined in Algorithm 2. The *MaximumJudge* function evaluates whether the child information set $I_p^{son}$ of $I_p$ has the highest expected value for player $p$. The *LegalFromRoot* function innovates further by verifying that player $p$ has not taken any sub-optimal actions from the initial state up to $I_p$. The *Pruning* function orchestrates the overall pruning process by using a queue to traverse the information sets and apply the pruning criteria. Ultimately,

the function returns the necessary information sets to retain, completing the efficient permanent pruning of the game tree before initiating the CFR algorithm.

---

**Algorithm 2:** Algorithm for EVPA subgame pruning.

---

**Function** `MaximumJudge`$(I_p, I_p^{son})$**:**

  **if** $\mathcal{P}(I_p) \neq p$ **then**

   **return** *True* // If the acting player is not $p$, return true

  $MaxMin \leftarrow -\infty$

  **for** $I_p' \in I_p \cdot a$ **do**

   **if** $I_p^{son} \neq I_p' \wedge EV_{min}(I_p') > MaxMin$ **then**

    $MaxMin \leftarrow EV_{min}(I_p')$

  **if** $EV_{max}(I_p^{son}) + \delta > Min(MaxMin, EV_{min}(I_p))$ **then**

   **return** *True*

  **return** *False*

**Function** `LegalFromRoot`$(I_p)$**:**

  **if** $Parent(I_p) = \emptyset$ **then**

   **return** *True*

  $I_p' \leftarrow Parent(I_p)$ // There exists an action $a$ such that $I_p' \cdot a = I_p$

  **return** $MaximumJudge(I_p', I_p) \wedge LegalFromRoot(I_p')$

**Function** `Pruning`$(S)$**:**

  $\mathcal{I}_{root}, \mathcal{I}_{subgame} \leftarrow \emptyset$

  **for** $I_p \in S_r$ **do**

   **if** $LegalFromRoot(I_p)$ **then**

    Add $I_p$ to $\mathcal{I}_{root}$ // Pruning root information sets

  $\mathcal{I}_{queue}, \mathcal{I}_{subgame} \leftarrow \mathcal{I}_{root}$

  **while** $\mathcal{I}_{queue} \neq \emptyset$ **do**

   $I_p \leftarrow TopElement(\mathcal{I}_{queue}), \mathcal{I}_{queue} \leftarrow \mathcal{I}_{queue} \setminus I_p$

   **for** $I_p' \in I_p \cdot a$ **do**

    **if** $s(I_p') \in S \wedge MaximumJudge(I_p, I_p')$ **then**

     Add $I_p'$ to $\mathcal{I}_{queue}, \mathcal{I}_{subgame}$

  **return** $\mathcal{I}_{root}, \mathcal{I}_{subgame}$

---

The parameter $\delta$ is introduced to mitigate errors generated by sampling data. When $\delta$ is no less than the maximum average regret $O(\frac{1}{\sqrt{T}})$ of CFR, where $T$ is the CFR iterations, we can eliminate the error generated by CFR with DeepStack.[3] After $\delta$ elimination of errors generated by CFR iterations in the sampling process, we can be confident that the sampled data fully reflects the exact expected value of the information set. For the information set $I_p$, the value estimation of the neural network can be expressed as $EV_{\sigma^*}(I_p) + err$, where $err$ is the random error generated by the network itself.

**Error Analysis.** We consider a pruning error to occur when a branch $I_p^{son} = I \cdot a$ with $EV_{\sigma^*}(I_p^{son}) = EV_{\sigma^*}(I)$ is pruned. In the worst-case scenario, all branches of the information set $I$ have the same expected value, making it incorrect to prune any branch. To analyze the probability of error, we consider the sampling of estimates from $M$ networks. We enumerate each sibling information set $I_p^{bro}$ of $I_p^{son}$, the pruning error occurs in the $I_p^{son}$ branch if all $M$ estimates about $I_p^{son}$ are smaller than the other $2M$ estimates about $I_p$ and the sibling information set $I_p^{bro}$. The probability of selecting exactly the $M$ minimum values out of $3M$ random values is $\frac{1}{\binom{3M}{M}}$. The $I_p^{son}$ has $|A| - 1$ siblings, and the upper bound pruning error probability of $I_p^{son}$ is $\frac{|A|-1}{\binom{3M}{M}}$.

The pruning algorithm estimates the expected values of $O(M \cdot D \cdot |\mathcal{I}| + M \cdot |S| \cdot |\mathcal{I}|)$ information sets, where $M$ represents the number of expected value estimation networks, $D$ is the current depth of the subgame, $|S|$ is the number of nodes in the subgame, and $|\mathcal{I}|$ denotes the number of distinct information sets per node. Notably, the computational overhead of the pruning algorithm is minimal compared to that of the CFR algorithm, highlighting its practicality and efficiency in large IIEFGs.

---

[3] The value of $\delta = 0.01$ was chosen based on empirical evidence. We observed that with this value, the error generated by discounted CFR (DCFR) algorithm (Brown & Sandholm, 2019b) can be effectively mitigated.

By directly and correctly eliminating unimportant branches, EVPA's pruning algorithm enables the CFR algorithm to concentrate on more critical branches, significantly enhancing convergence and overall performance. This strategic focus not only accelerates computation but also facilitates more effective exploration of promising areas within the game tree.

### 4.3 Information Abstraction for Subgames

In this subsection, we describe the Information Abstraction for Subgames component of the EVPA algorithm. This crucial part of our approach aims to simplify the game tree by merging similar information sets, thus reducing computational complexity while maintaining strategic integrity.

EVPA's information abstraction method innovatively clusters both current and future expected values of information sets as features. The core idea behind merging information sets $I$ and $I'$ is to simplify the game tree by grouping similar information sets together. We consider them similar if their current and future expected values are closely aligned. Specifically, we merge two information sets $I$ and $I'$ when the following conditions are met: (1) $s(I) = s(I')$, which ensures that the public states of the two information sets are the same, and (2) for any action sequences $a_1, \cdots, a_n$ (where $n \geq 0$), if $\forall i < n, \mathcal{P}(I \cdot a_1 \cdots a_i) \neq \mathcal{P}(I)$, then $EV_{\sigma^*}(I \cdot a_1 \cdots a_n) \approx EV_{\sigma^*}(I' \cdot a_1 \cdots a_n)$. This second condition checks that for any sequence of actions, the expected values of both information sets remain approximately equal. If these conditions hold, it follows that $\sigma^*(I) \approx \sigma^*(I')$, indicating that the strategies for $I$ and $I'$ will be similar, and hence they can be merged.

This merging approach extends to subgame solving, allowing for the consolidation of root information sets into $K$ buckets within subgames. For player $p$, all information sets in bucket $k$ are denoted as $\mathcal{I}_{rootbucket,p,k}$. All information sets in $\mathcal{I}_{rootbucket,p,k}$ adopt the same strategy in the subgame.

The details of the EVPA information abstraction algorithm are outlined in Algorithm 3. The process begins by pruning all root information sets and storing the viable root information sets in $\mathcal{I}_{root}$.

Next, we enumerate each node $s$ of the subgame $S$ and predict the expected value of each root information set $I_p$ at that node. To estimate this expected value, we utilize the average output from $M$ expected value estimation networks. This estimated value is then multiplied by a significance function $f(g(s), n)$ and appended to the features of $I_p$. Here, $g(s)$ encapsulates various pieces of information about node $s$, while $n$ represents the distance from the root node $S_r$ to node $s$. In the context of HUNL, we define the significance function as $f(g(s), n) = \frac{1}{pot(s) \cdot \max\{n, 0.2\}}$, where $pot(s)$ indicates the number of chips in the pot at node $s$.

---

**Algorithm 3:** Algorithm for EVPA information abstraction.

**Function** `Abstraction(`$S, K$`):`
  $\mathcal{I}_{root}, \mathcal{I}_{subgame} \leftarrow Pruning(S)$
  **for** $I_p \in \mathcal{I}_{root}$ **do**
    | $feature(I_p) \leftarrow \emptyset$ // Define features for information sets
  **for** $s \in S$ **do**
    | $a_1, \cdots, a_n \leftarrow SequenceActions(S_r, s)$ // Actions from $S_r$ to $s$
    | **for** $I_p \in \mathcal{I}_{root}$ **do**
      | Append $\overline{EV}(I_p \cdot a_1 \cdots a_n) \cdot f(g(s), n)$ into $feature(I_p)$ // Append weighted average expected value to the feature vector of $I_p$
  $\mathcal{I}_{rootbucket}, \mathcal{I}_{subgamebucket} \leftarrow \emptyset$
  **for** $p \in \mathcal{N}$ **do**
    | $\mathcal{I}_{rootbucket,p,1}, \ldots, \mathcal{I}_{rootbucket,p,K} \leftarrow k\text{-means++}(\{I_p, feature(I_p)\}_{I_p \in \mathcal{I}_{root}}, K)$
    | **for** $s \in S$ **do**
      | $a_1, \cdots, a_n \leftarrow SequenceActions(S_r, s)$
      | **for** $k = 1$ *to* $K$ **do**
        | **for** $I_p \in \mathcal{I}_{rootbucket,p,k}$ **do**
          | **if** $I_p \cdot a_1 \cdots a_n \in \mathcal{I}_{subgame}$ **then**
            | Add $\mathcal{I}_{rootbucket,p,k}$ to $\mathcal{I}_{subgamebucket,p,s}$
            | **break** // Exit loop once a match is found
  **return** $\mathcal{I}_{rootbucket}, \mathcal{I}_{subgamebucket}$

---

After calculating expected value features, we cluster the root information sets for each player using the $k$-means++ algorithm (Arthur & Vassilvitskii, 2007). The clustering process has a time complexity of $O(T \cdot K \cdot |S| \cdot |\mathcal{I}_{root}|)$, where $T$ is the number of iterations of the $k$-means++ algorithm (set to $T = 8$), $K$ is the number of buckets, $|S|$ is the number of nodes in the subgame, and $|\mathcal{I}_{root}|$ is the number of legitimate information sets at the root node. Remarkably, even for large IIEFGs such as HUNL, the information abstraction process can be completed quickly, making it suitable for real-time applications.

Following the clustering, we retain the original pruning results of the subgame in $\mathcal{I}_{subgame}$. After clustering, we check for buckets where all of their information sets are not in the previously pruned subgame set $\mathcal{I}_{subgame}$. These buckets are considered unnecessary and can be pruned, further reducing the complexity of the game tree.

A key innovation of EVPA is its ability to provide dynamic, subgame-specific abstractions that effectively integrate both current and future expected values. Compared to other information abstraction methods, EVPA's approach not only offers faster processing times (completing in under 1 second on a standard server for HUNL) but also provides more accurate and dynamic abstractions tailored to specific subgames, as it takes into account both current and future expected values. Additionally, the flexibility in choosing the bucket size $K$ allows for an optimal balance between solving time and abstraction accuracy, further enhancing the algorithm's applicability in real-time scenarios.

## 5 EXPERIMENTS

We conduct a series of experiments to comprehensively evaluate the performance of the EVPA algorithm in the context of large IIEFGs. As in previous studies on large IIEFGs (Moravčík et al., 2017; Brown & Sandholm, 2018), we use HUNL (see Appendix A for detailed rules) as our experimental benchmark due to its representativeness and complexity. In our evaluation, players start with 200 big blinds and switch positions every two hands, replicating conditions of the annual computer poker competition (ACPC) (Bard et al., 2013). All experiments employed the leading DCFR algorithm (Brown & Sandholm, 2019b), as outlined in Appendix C. To maintain strategy soundness, we utilized subgame re-solving techniques (Burch et al., 2014; Brown & Sandholm, 2017c). Additionally, the AIVAT technique (Burch et al., 2018) is applied to reduce variance in heads-up evaluations.

Training and experiments were executed on a server with 4 NVIDIA A100 80GB PCIe GPUs and 112 Intel(R) Xeon(R) Gold 6348 2.60GHz CPUs. For our evaluations, we replicated DeepStack (Moravčík et al., 2017) as a baseline (BASE) and implemented three AIs using the EVPA algorithm: EVPA-full (which employs pruning only), EVPA-169 (pruning combined with 169 buckets for abstraction), and EVPA-30 (pruning combined with 30 buckets for abstraction).

**Pruning Effectiveness.** We randomly generated at least $10,000$ depth-limited subgames across different stages of HUNL. The pruning rates are summarized in Table 1. EVPA-full achieved pruning rates between $69.72\%$ and $79.51\%$. EVPA-169 achieved pruning rates between $55.68\%$ and $69.50\%$, while EVPA-30 achieved pruning rates between $42.67\%$ and $51.58\%$. These results clearly indicate that EVPA effectively prunes a significant portion of the game tree across various settings.

Table 1: Game tree pruning rate of depth-limited subgame at each stage of HUNL.

| EVPA AIs | Pre-flop | Flop | Turn | River |
|----------|----------|------|------|-------|
| EVPA-full | 69.72% | 69.87% | 77.00% | 79.51% |
| EVPA-169 | 69.50% | 55.68% | 63.07% | 65.31% |
| EVPA-30 | 51.58% | 42.67% | 49.60% | 49.79% |

**Exploitability Evaluation.** In our evaluation of exploitability at the river stage subgame, as illustrated in Figure 3, the EVPA AIs demonstrate a marked reduction in exploitability compared to the baseline. Specifically, EVPA-full achieves the lowest exploitability with $1 \times 10^8$ information set touches, reducing exploitability to approximately $1.4\%$ of the baseline's level. Further analysis reveals that EVPA-169 reduces exploitability by 47% relative to EVPA-full with $1 \times 10^7$ touches, while EVPA-30 achieves a $54\%$ reduction relative to EVPA-169 with $1 \times 10^6$ touches.

Notably, to reach an exploitability of no more than 0.08, the baseline requires around $1 \times 10^8$ information set touches, whereas EVPA-30 achieves this with only $1 \times 10^6$. Similarly, for an exploitability threshold of 0.01, the baseline necessitates over $5 \times 10^8$ touches, while EVPA-169 can already achieve this with just $1 \times 10^7$ touches. These results underscore that EVPA can attain an $\varepsilon$-Nash equilibrium with only 1%-2% of the information set touches required by the baseline, highlighting the significant improvement in exploitability reduction achieved by EVPA.

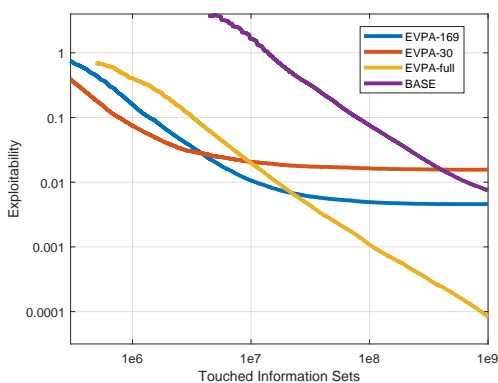

Figure 3: The average exploitability (relative to the current pot size) of various HUNL AIs during river subgames is plotted against the number of information sets touched by DCFR.

**Heads-up Evaluation Against Slumbot.** We conducted heads-up evaluations against Slumbot (Jackson, 2013), a strong open-source HUNL AI that won the 2018 ACPC. The results are presented in Table 2. In the initial trials with a solving time limit of 0.02 seconds, DeepStack's replication suffered a significant defeat, recording a win-rate of $-573 \pm 28$ mbb/h against Slumbot. In contrast, the EVPA algorithms—EVPA-full, EVPA-169, and EVPA-30—demonstrated comparable performance levels to Slumbot, with EVPA-169 achieving a win-rate of $10 \pm 26$ mbb/h. When the solving time was increased to 0.2 seconds, DeepStack's replication continued to struggle, showing a win-rate of $-109 \pm 52$ mbb/h. Meanwhile, EVPA-full succeeded in defeating Slumbot with a win-rate of $96 \pm 43$ mbb/h. When the solving time was further increased to 2 seconds, DeepStack's replication managed to beat Slumbot with a win-rate of $33 \pm 65$ mbb/h, but still fell short of EVPA's performance. Overall, these results show that EVPA algorithms are more competitive against Slumbot, especially under time-constrained scenarios.

Table 2: Heads-up results of AIs against Slumbot with solving time limits, measured in mbb/h.

| Solving Time Limits | BASE | EVPA-full | EVPA-169 | EVPA-30 |
|---|---|---|---|---|
| 0.02 seconds | $-573 \pm 28$ | $8 \pm 25$ | $10 \pm 26$ | $-4 \pm 28$ |
| 0.2 seconds | $-109 \pm 52$ | $96 \pm 43$ | $31 \pm 57$ | - |
| 2 seconds | $33 \pm 65$ | $100 \pm 68$ | - | - |

**Heads-up Evaluation Against DeepStack's Replication.** We further conducted heads-up evaluation between DeepStack's replication and various EVPA AIs, varying the limits on the number of information set touches, as presented in Table 3. After the CFR algorithm touches the information set $1 \times 10^7$ times, the EVPA algorithm significantly outperforms DeepStack's replication, demonstrating a win-rate of up to $930 \pm 23$ mbb/h. Even after $1 \times 10^8$ touches, EVPA maintained its advantage, defeating DeepStack's replication with a win-rate of $202 \pm 31$ mbb/h. When the number of touched information sets came to $1 \times 10^9$, DeepStack's replication was still outperformed by EVPA, with recorded a win-rate of $82 \pm 60$ mbb/h. These results show that EVPA has a significant win-rate against DeepStack's replication, regardless of the number of information set touches.

Table 3: Heads-up results of EVPA AIs against DeepStack's replication with different limits on the number of information set touches, measured in mbb/h.

| Touched Information Sets | EVPA-full | EVPA-169 | EVPA-30 |
|---|---|---|---|
| $1 \times 10^7$ | $844 \pm 26$ | $930 \pm 23$ | $865 \pm 29$ |
| $1 \times 10^8$ | $202 \pm 31$ | $125 \pm 27$ | - |
| $1 \times 10^9$ | $82 \pm 60$ | - | - |

Overall, the experimental results show that EVPA is a highly effective algorithm for solving large IIEFGs, particularly in settings with limited solving time.

ACKNOWLEDGEMENTS

This work was supported by the National Natural Science Foundation of China Grants 52450016 and 52494974.

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

# Appendix

## A    HEADS-UP NO-LIMIT TEXAS HOLD'EM RULES

Heads-up No-Limit Texas Hold'em (HUNL) is a two-player variant of Texas Hold'em poker, played over four stages:

1. *Pre-flop*. Players start by posting a pre-specified number of chips: the "small blind" and the "big blind," with the small blind typically being half the size of the big blind. Each player is dealt two private cards at the beginning.

2. *Flop*. Three public cards are revealed.

3. *Turn*. A fourth public card is revealed.

4. *River*. The final public card is revealed.

During the pre-flop stage, the small blind player acts first; thereafter, the big blind player acts first in all subsequent stages. Players can fold, check/call, or bet/raise, with bets/raises ranging from the last bet/raise amount to their remaining chips (all-in). If a player folds, the other wins the pot. If neither folds by the end of the river stage, the players compare their best five-card hands, which consist of two private cards and the five public cards. The player with the best hand wins the pot. Win-rate and exploitability are measured in milli big blinds per hand (mbb/h). For instance, a win-rate of 0.1 big blind per hand equates to 100 mbb/h.

## B    BREAKTHROUGHS IN HUNL AIS

Tartanian7 (Brown et al., 2015), Baby Tartanian8 (Brown & Sandholm, 2016b), and Slumbot (Jackson, 2013) — winners of annual computer poker competition (ACPC) (Bard et al., 2013) in 2014, 2016, and 2018, respectively — initially computed a blueprint strategy based on a post-abstraction game tree (Ganzfried & Sandholm, 2014; Brown & Sandholm, 2014), followed by the strategy execution based on the blueprint strategy (Ganzfried et al., 2011; 2012; Ganzfried & Sandholm, 2013a; Brown & Sandholm, 2016a). While this approach is effective for IIEFGs with fewer states, such as limit Texas Hold'em (Tammelin et al., 2015), it struggles in HUNL. A significant challenge arises when an opponent makes an "off-tree" action, in such cases, the AI must substitute an approximation from the blueprint strategy (Schnizlein et al., 2009; Ganzfried & Sandholm, 2013a), potentially leading to sub-optimal performance (Ganzfried & Sandholm, 2013b; Bard et al., 2014). Additionally, the blueprint strategy often lacks the granularity necessary for effective endgame solving (Ganzfried & Sandholm, 2013b; 2015).

Libratus (Brown & Sandholm, 2018) addressed these challenges through a safe and nested sub-game solving technique (Brown & Sandholm, 2017c) and a self-improver to enhance its blueprint strategy (Brown & Sandholm, 2017b), ultimately achieving superhuman performance. However, its substantial computational resource requirements limit its implementation to supercomputers, and it can only solve for fixed initial chip counts, which is impractical in real poker scenarios where chip counts fluctuate (Burch et al., 2018).

DeepStack (Moravčík et al., 2017) achieved superhuman performance through depth-limited sub-game solving (Kroer & Sandholm, 2015) and probability distribution-based value estimation of leaf information sets. Subsequent works, such as ReBeL (Brown et al., 2020) and Supremus (Zarick et al., 2020), further demonstrate the reliability of this approach. However, none of these AIs have incorporated online pruning and abstraction of depth-limited subgames, suggesting potential areas for improvement. Modicum (Brown et al., 2018) introduced a multiple-value depth-limited solving technique based on a pre-calculated blueprint strategy, which allows for minimal training and solving time. Nevertheless, its performance remains constrained by the limitations of the blueprint strategy. AlphaHoldem (Zhao et al., 2022), a reinforcement learning-based method, achieves decent performance with short training and solving times. However, its lack of CFR integration makes it susceptible to exploitation.

Table 4 compares the training and solving time of these AIs and the EVPA algorithm, along with their win-rates against Slumbot or Baby Tartanian8. EVPA demonstrates significant advantages in processing off-tree actions compared to blueprint strategy-based AIs, achieving higher win-rates than these methods. Additionally, EVPA outperforms other subgame-solving AIs in solving time while maintaining a comparable win-rate. Though it has similar solving times and win-rates to AlphaHoldem, EVPA provides stronger guarantees against exploitability. [4]

Table 4: This table compares various HUNL AIs based on estimated training and solving times using 4 NVIDIA A100 80GB PCIe GPUs and 112 Intel(R) Xeon(R) Gold 6348 CPUs. Win-rates indicate performance against Slumbot or Baby Tartanian8, measured in mbb/h. Note that Slumbot is continually improved, so earlier results may not reflect its current performance. $X$ is the training days of the baseline strategy.

| HUNL AI Name | Training Time | Solving Time | Win-rate |
|---|---|---|---|
| Tartanian7 (Brown et al., 2015) | 428 Days | 0 seconds | $33 \pm 16$ |
| Baby Tartanian8 (Brown & Sandholm, 2016b) | 744 Days | 0 seconds | $36 \pm 12$ |
| Slumbot (Jackson, 2013) | 93 Days | 0 seconds | $0 \pm 0$ |
| DeepStack (Moravčík et al., 2017) | 569 Days | 2 seconds | - |
| Libratus (Brown & Sandholm, 2018) | 8,370 Days | 160 seconds | $63 \pm 28$ |
| Modicum (Brown et al., 2018) | $X + 1$ Days | 2 seconds | $11 \pm 9$ |
| ReBeL (Brown et al., 2020) | 926 Days | 1 second | $45 \pm 5$ |
| Supremus (Zarick et al., 2020) | 351 Days | 2 seconds | $176 \pm 44$ |
| AlphaHoldem (Zhao et al., 2022) | 6 Days | 0.003 seconds | $112 \pm 16$ |
| EVPA-full-0.2s (**Ours**) | $X + 4$ Days | 0.2 seconds | $96 \pm 43$ |
| EVPA-full-0.2s-confuseSlumbot (**Ours**) | $X + 4$ Days | 0.2 seconds | $187 \pm 66$ |
| EVPA-169-0.02s (**Ours**) | $X + 4$ Days | 0.02 seconds | $10 \pm 26$ |

## C  COUNTERFACTUAL REGRET MINIMIZATION

Counterfactual Regret Minimization (CFR) is a prominent algorithm for solving large IIEFGs by minimizing regret independently at each information set (Zinkevich et al., 2007b). CFR can find $\varepsilon$-Nash equilibrium in two-player zero-sum IIEFGs.

---

[4]Notably, Supremus and AlphaHoldem often make decisions that deviate from Slumbot's blueprint strategy, which may confuse Slumbot. For a fairer comparison with Supremus and AlphaHoldem, we implemented an EVPA-based AI using bet/raise sizes of 0.2, 0.4, 0.8, 1.6 times the pot (EVPA-full-0.2s-confuseSlumbot). The results show that EVPA-full-0.2s-confuseSlumbot outperformed Slumbot, achieving a win-rate of $187 \pm 66$ mbb/h, surpassing both Supremus and AlphaHoldem.

$\pi^{\sigma}(h)$ is the probability of reaching $h$ if all players act according to strategy $\sigma$. $\pi^{\sigma}_{-p}(h)$ is the probability of reaching $h$ if all players expect $p$ act according to strategy $\sigma$, and the player $p$ act to $h$. The counterfactual value (CFV) of an information set $I$ under strategy profile $\sigma$ represents the expected utility to player $\mathcal{P}(I)$ if $I$ has been reached, calculated as $v^{\sigma}(I) = \sum_{h \in I}(\pi^{\sigma}_{-\mathcal{P}(I)}(h|I)\sum_{h \sqsubseteq z}(\pi^{\sigma}(z|h)u_{\mathcal{P}(I)}(z)))$. And the counterfactual value of an action $a$ is calculated as $v^{\sigma}(I, a) = \sum_{h \in I}(\pi^{\sigma}_{-\mathcal{P}(I)}(h|I)\sum_{h \sqsubseteq z}\pi^{\sigma}(z|h \cdot a)u_{\mathcal{P}(I)}(z))$.

Let $\sigma^t$ denote the strategy profile at iteration $t$. The instantaneous regret for taking an action $a$ at information set $I$ during iteration $t$ is given by: $r^t(I, a) = v^{\sigma^t}(I, a) - v_p^{\sigma^t}(I)$, where $v^{\sigma^t}(I, a)$ is the counterfactual value of taking an action $a$ at $I$, and $v^{\sigma^t}(I)$ is the counterfactual value of the information set $I$. The accumulated counterfactual regret for taking an action $a$ at information set $I$ after $T$ iterations is: $R^T(I, a) = \sum_{t=1}^{T} r^t(I, a)$. At each iteration $t$, an action $a$ at information set $I$ is selected with probability: $\sigma^t(I, a) = \frac{R_+^{t-1}(I,a)}{\sum_{a'} R_+^{t-1}(I,a')}$, where $R_+^t(I, a) = \max\{0, R^t(I, a)\}$.[5]

Discounted CFR (DCFR) (Brown & Sandholm, 2019b) is an advanced variant of CFR designed for large IIEFGs. DCFR introduces parameters $\alpha, \beta$ and $\gamma$ to adjust the impact of accumulated counterfactual regrets over time.[6] Specifically, at each iteration $t$, positive accumulated counterfactual regrets are multiplied by $\frac{t^{\alpha}}{t^{\alpha}+1}$, negative accumulated counterfactual regrets are multiplied by $\frac{t^{\beta}}{t^{\beta}+1}$, and contributions to the average strategy $\overline{\sigma}$ are weighted by $(\frac{t}{t+1})^{\gamma}$.

## D   DEEPSTACK'S IMPLEMENTATION

Our replication of DeepStack's implementation builds upon several prior works (Moravčík et al., 2017; Zarick et al., 2020; Brown et al., 2020). At each stage of the game, we construct a depth-limited subgame up to the end of that stage and utilize a neural network to estimate the values of the leaf information sets (excluding the river stage).

For the first two levels of the subgame, we employ raising scales of $0.25$, $0.5$, $1$, and $2$ times the pot, along with an all-in option. For the third level, we use scales of $0.5$ and $1$ times the pot, plus all-in. For subsequent levels, we apply a raising scale of $0.75$ times the pot and an all-in option. To enhance sample diversity, we randomly multiply all raising scales (except for all-in) by a factor between $0.7$ and $1.4$. [7]

We train 6 neural networks corresponding to the following stages: pre-flop stage end, flop stage start, flop stage end, turn stage start, turn stage end, and river stage start. Each network consists of 6 layers of Multi-Layer Perceptrons (MLPs) with ReLU activation functions (Glorot et al., 2011). The networks are trained using the Adam optimizer (Kingma & Ba, 2015) and the Huber loss function (Huber, 1964). The input layer has $2,678$ dimensions, corresponding to the probability of private hands for both players and the public state information. Each hidden layer contains $1,536$ dimensions, while the output layer has $2,652$ dimensions, representing the expected value of the private hands.

Training progresses as follows: we first train the river stage network with 5 million randomly generated river subgames, followed by training the turn stage network with 3 million randomly generated turn subgames, the flop stage network with 1 million randomly generated flop stage subgames, and finally the pre-flop stage network with $100,000$ randomly generated pre-flop scenarios. After this initial training, we regenerate samples from initial states (with initial chips between 50 and 250 big blinds) to terminal states for 100 epochs, generating at least 36 million samples in total. After each epoch, we retrain the neural networks using the most recent samples. In the EVPA process for sampling and evaluation, we apply the depth-limited subgame building method and leverage the trained neural networks to estimate the values of the leaf information sets.

It is important to note that DeepStack dedicates a significant amount of training time to computing the information abstraction due to its reliance on offline abstraction methods. To reduce training overhead, our implementation of the DeepStack replica omits this information abstraction step and

---

[5]If $\sum_{a'} R_+^t(I, a') = 0$, an arbitrary strategy is used.

[6]In our experimental setup, we use $\alpha = 1.5$, $\beta = 0$ and $\gamma = 2$.

[7]During sample generation, we use the DCFR algorithm (Brown & Sandholm, 2019b) for $1,000$ iterations.

does not fix the initial chip count. Furthermore, inspired by methods such as Supremus (Zarick et al., 2020) and ReBeL (Brown et al., 2020), we incorporate a warm-up phase and then the trained neural network is used to generate subsequent data, allowing for more efficient training process.

## E    Conclusion

In this paper, we propose the Expected-Value Pruning and Abstraction (EVPA) algorithm, which marks a significant advancement in solving large imperfect information extensive-form games. EVPA is featured with three core components, namely expected value estimation of information sets, expected value-based pruning, and information abstraction for subgames. Our extensive experiments with Heads-up No-Limit Texas Hold'em (HUNL) show that EVPA enhances computational efficiency while ensuring robust strategy development, achieving competitive performance with significantly reduced solving time compared to existing benchmarks. Its dynamic adaptability positions EVPA as a pivotal tool for advancing AI capabilities in complex strategic environments.

## F    Future Directions for Research and Application

As EVPA demonstrates significant advancements in solving HUNL, several promising directions for future research emerge.

- *Generalization to Other IIEFGs.* A compelling avenue for future research is to explore the applicability of EVPA across a broader spectrum of IIEFGs. While the methodology has been validated in HUNL, expanding its scope to games such as Omaha (Farha & Reback, 2007), Mahjong (Li et al., 2020) and Stratego (Perolat et al., 2022) could yield insights into the versatility of the EVPA framework. Understanding how EVPA can efficiently manage diverse information structures and player strategies in these contexts will contribute to more generalized solutions in game theory.

- *Online Strategy Solving Beyond Board Games.* The development of EVPA positions it well for applications beyond board games, particularly in online strategy solving. Many real-world scenarios involve sequential decision-making under uncertainty, such as auctions (Milgrom & Weber, 1982) and diplomacy (Bakhtin et al., 2023). Future work could focus on adapting the EVPA framework to these domains, enabling robust online strategy solving that leverages its expected value-based pruning and abstraction techniques. Implementing EVPA in these settings could provide a powerful tool for developing competitive agents capable of navigating complex interactions and dynamic information.

- *Integration with Multi-Agent Systems.* EVPA could potentially solve multi-player poker, particularly in tournament settings (Snyder, 2006; Ganzfried & Sandholm, 2009), in mere seconds. This capability provides a promising entry point for exploring EVPA's application in multi-agent systems. By expanding EVPA's functionalities to operate effectively in both collaborative and adversarial environments, we can investigate its adaptability to scenarios involving multiple agents, each with unique strategies and objectives. This exploration will facilitate more sophisticated strategic interactions, proving particularly valuable in domains such as negotiations (Thompson, 1990) and cooperative games (Driessen, 2013).

By pursuing these research directions, future work can build on the successes of EVPA, enabling the development of robust and adaptable strategies for a wide range of sequential decision-making problems beyond poker. This expansion will contribute to the ongoing evolution of AI in complex strategic environments, pushing the boundaries of what is achievable in the field of IIEFGs.

## G    Pruning with CFR

Pruning techniques in CFR help the algorithm avoid exploring suboptimal branches of the game tree, thus improving computational efficiency without sacrificing convergence guarantees (Brown & Sandholm, 2015a). One of the most widely used pruning methods is partial pruning (Lanctot et al., 2009)[8], which reduces unnecessary computations when updating the regret of an information set

---

[8]Dynamic thresholding pruning (Brown et al., 2017) is similar to partial pruning in CFR.

belonging to one player. Specifically, if the other player has zero probability of reaching any history within an information set, the subgame rooted at that history can be pruned without affecting the overall computation. Formally, for an information set $I$ and the strategy profile $\sigma^t$ at iteration $t$, if history $h \in I$ satisfies $\pi^{\sigma^t}_{-\mathcal{P}(I)}(h) = 0$, the subgame rooted at $h$ can be pruned in iteration $t$. This technique can be combined with EVPA or other pruning methods for even more effective pruning during CFR iterations. However partial pruning techniques are unable to prune the information set $I$ belonging to the action player $\mathcal{P}(I)$, limiting its pruning rate.

Regret-based pruning (RBP) (Brown & Sandholm, 2015a) avoids traversing branches where either player is unlikely to take actions with positive probability. If the cumulative counterfactual regret $R(I, a) \leq 0$ for an action $a$ in information set $I$, RBP temporarily prunes the path from $I$ to $a$ for $\lfloor \frac{-R(I,a)}{U(I,a)-L(I)} \rfloor$ iterations, where $U(I, a)$ is the upper bound of $v(I, a)$ and $L(I)$ is the lower bound of $v(I)$. Although RBP is efficient for standard CFR, many CFR variants (Brown & Sandholm, 2019b; Xu et al., 2024) require more complicated computation of $R(I, a)$, as it is not always a simple cumulative quantity. Additionally, RBP necessitates computing both the upper and lower bounds and recalculating the best response after pruning. These extra computations introduce significant overhead, particularly in real-time solving scenarios.

Best-Response Pruning (BRP) (Brown & Sandholm, 2017a) is another approach that eliminates suboptimal branches, based on the assumption that players will avoid suboptimal actions. For each information set $I$ and action $a$, BRP computes the best response against the player $-\mathcal{P}(I)$ in a subgame of non-suboptimal actions. A regret upper bound $U$ for information set $I$ and action $a$ with $T$ iterations is derived. If $U < 0$, the path from $I$ to $a$ can be pruned for the next $\lfloor \frac{U}{L(I)} \rfloor$ iterations. While BRP can improve both convergence speed and computational efficiency, it introduces complexity by requiring the computation of the best response at each information set during every iteration. This is particularly challenging in depth-limited solving, where best responses calculation is costly compared to the depth-limited subgame solving (Moravčík et al., 2017).

Table 5: Comparison of Pruning Methods with CFR.

| Pruning Methods | Depth-limited Solving | Permanent Pruning |
|---|:---:|:---:|
| Partial Pruning (Lanctot et al., 2009) | ✔ | ✘ |
| RBP (Brown & Sandholm, 2015a) | ✘ | ✘ |
| BRP (Brown & Sandholm, 2017a) | ✘ | ✘ |
| EVPA(**Ours**) | ✔ | ✔ |

The comparison of EVPA with previous pruning algorithms is summarized in Table 5. The key difference between EVPA and other pruning methods is that EVPA offers permanent pruning, making it compatible with techniques such as depth-limited solving. Furthermore, EVPA does not require any additional overhead for intermediate value calculations after pruning is completed. This makes EVPA not only more efficient but also more versatile, as it can be seamlessly integrated with various CFR variants. As a result, EVPA offers substantial performance improvements without the added computational burden associated with methods like RBP or BRP.

# H  ABSTRACTION FOR SOLVING LARGE IIEFGS

In large IIEFGs, the size of the game tree often makes it computationally prohibitive to compute an equilibrium. Abstraction algorithms help address this challenge by simplifying the original game into a smaller, abstracted version, which is then solved for equilibrium strategies. While abstraction techniques necessarily sacrifice some accuracy, they are theoretically bounded (Kroer & Sandholm, 2016). However, determining the optimal abstraction remains an NP-complete problem (Sandholm & Singh, 2012; Kroer & Sandholm, 2014).

Abstraction algorithms typically reduce the complexity of the game tree in the following ways: 1. **Information Abstraction** (Gilpin & Sandholm, 2007; Ganzfried & Sandholm, 2014): This method groups similar information sets together, treating them as a single entity with a shared strategy. 2. **Action Abstraction** (Brown & Sandholm, 2014; 2015b; Li et al., 2024): By limiting the available

actions in the game, action abstraction reduces the size of the game tree. 3. **Depth-Limited Solving** (Kroer & Sandholm, 2015; Moravčík et al., 2017; Brown et al., 2018): This approach limits the depth to which the game tree is explored, reducing the computational load. 4. **Subgame Solving** (Ganzfried & Sandholm, 2013b; 2015; Brown & Sandholm, 2017c): Instead of solving the entire game tree, subgame solving focuses on specific portions of the tree, starting from the current state, to avoid unnecessary computation.

In EVPA, we leverage advanced techniques such as action abstraction, depth-limited solving, and subgame solving simultaneously, while also introducing a novel approach to information abstraction. This combination enables us to achieve substantial improvements in efficiency and performance. The key advantages of EVPA include: 1. **Online Execution**: EVPA performs the abstraction in less than 1 second for HUNL subgames, making it highly efficient in real-time applications. 2. **Efficient Performance**: EVPA excels in abstraction quality, providing significant performances in convergence speed and eventual exploitability. When combined with subgame solving techniques, EVPA's abstraction method has more significant advantages over other offline abstraction methods.

