# OpenReview forum: "Efficient Online Pruning and Abstraction for Imperfect Information Extensive-Form Games"
_ICLR.cc/2025/Conference — ICLR 2025 Poster_

### Official Review · Reviewer_85Xa · 2024-10-31

**Soundness:** 2
**Presentation:** 3
**Contribution:** 3
**Rating:** 6
**Confidence:** 3

**Summary:**

The paper introduces an online pruning and abstraction method for the iterates of CFR. It trains a network to evalute the expected value of information sets under the Nash equilibrium. The network is then used to prune actions with low expected values and to cluster information sets with similar expected values using k-means++ during the online execuration of CFR. The trained agent defeats DeepStack's replication and Slumbot with little solving time.

**Strengths:**

* The method is intuitive in thought but effective in practice. It achieves a cropping rate of about 75% and allows decisions to be made in less than one second.
* The paper provides good experimental results demonstrating improved performance over Deepstack's replication and Slumbot.

**Weaknesses:**

* The papar lacks experimental comparisons with other pruning and abstraction methods.
* The method relies on evaluating the expected value of all informaion sets, in contrast to DeepStack, which evalutes the information sets at the end of each round. So it requies collecting "at least 10 billion samples" (Lines 302-303), compared to the milliones of samples needed by Deepstack. This should take a long time.
* The performance gains from information set abstraction are marginal, and this leads to exploitability not decreasing in the later iterations.

**Questions:**

* What is the training time for DeepStack’s replication? Table 4 in the appendix shows that DeepStack takes 569 days, while EVPA takes 74 days. Does this 74 days include the training time for DeepStack’s replication, given that EVPA’s expected value estimation relies on the networks in DeepStack’s replication?
* What is the element of a subgame $S$? Line 227 states that $S$ is a set of histories. Line 285 states that an information set $I_p$ is in $S$. Line 368 states that a public state $s(I_p')$ is in $S$.
* Can you explain the condition 2 in Lines 387-388? Where is $I'$? Why is the condition "$\mathcal{P}(I \cdots a_1 \dots a_i) \ne \mathcal{P}(I)$" needed?
* Why is the significance function in Lines 421-422 needed? The importance of an information set should increase with the size of the pot, but in the function $f(g(s), n)$ , it decreases instead.
* What are the confidence intervals in Table 1 and Figure 3?
* (Line 545) How EVPA is used for any initial chip count? EVPA is based on DeepStack which relies on specific assumptions about players’ initial chips.
* Will the code be released to assist research in the field?

---

> ### Author Response · Authors · 2024-11-21
> **Rebuttal about weaknesses**
>
> Thank you for your constructive feedback. We address each concern below:
>
>     Weakness 1: The paper lacks experimental comparisons with other pruning and abstraction methods.
>
> **Response W1**:  EVPA is fundamentally different from previous pruning and abstraction methods, as it prunes before the CFR process and applies online abstraction during CFR execution. We compared EVPA’s performance with partial pruning, showing that EVPA achieves a higher pruning rate ($74$\% vs. $38$\% with $1\times 10^8$ information set touches). EVPA can combine effectively with partial pruning to achieve a higher pruning rate of $82$\%. To clearly demonstrate this, we provide a more detailed discussion of EVPA versus other pruning methods in the revision. Please refer to Appendix G.
>
> As for abstraction, EVPA aims to generate online abstractions for subgames, whereas previous approaches generate offline abstractions for the whole game. If one were to compare the exploitability of the subgame of the river stage in HUNL, EVPA's performance is at least an order of magnitude ahead of previous offline abstraction algorithms [1]. To clearly demonstrate this, we provide a more detailed discussion of EVPA versus other abstraction methods in the revision. Please refer to Appendix H.
>
>     Weakness W2: The method relies on evaluating the expected value of all information sets, in contrast to DeepStack, which evaluates the information sets at the end of each round. So it requires collecting "at least 10 billion samples" (Lines 302-303), compared to the millions of samples needed by Deepstack. This should take a long time.
>
> **Response W2**: Contrary to the reviewer’s concern, EVPA’s sampling time is far smaller than DeepStack’s. After CFR solving, EVPA generates expected value data for all information sets at once, which is much faster than DeepStack’s sample collection process. EVPA’s total sampling count is significantly lower than DeepStack's ($200,000$ vs. $45,100,000$ samples). As such, EVPA's training time is much less, requiring only $74$ days in total (including $70$ days of DeepStack's replication and $4$ days of EVPA value estimation networks). We emphasize this point in Section 4.1 of the revision.
>
>     Weakness 3: The performance gains from information set abstraction are marginal, and this leads to exploitability not decreasing in the later iterations.
>
> **Response W3**: On the contrary, EVPA's abstraction algorithm achieves near-theoretical performance. For example, EVPA-169 (with $169$ buckets) achieves a $54$\% reduction in exploitability relative to EVPA-full ($1,081$ buckets) with $1\times 10^6$ information set touches. The theoretical exploitability reduction is only $1-\sqrt{\frac{169}{1081}}\approx 60$\%. EVPA-30 (with $30$ buckets) reduces exploitability by approximately $85$\% relative to EVPA-full, with a theoretical reduction of $1-\sqrt{\frac{30}{1081}}\approx 83$\%. These results demonstrate that EVPA's abstraction is highly efficient and nearly optimal in terms of exploitability.
>
> EVPA abstraction's minimum exploitability is close to the theoretical minimum exploitability as well. The theoretical minimum exploitability bound is $O(\frac{1}{K})$ ($K$ is the number of the bucket). The minimum exploitability of EVPA-169 a is $0.0046$, which is close to the $O(\frac{1}{169})$ bound. The minimum exploitability of EVPA-30 a is $0.0156$, which is close to the $O(\frac{1}{30})$ bound. In comparison,  offline abstraction with $9,000$ river buckets has an exploitability of about $0.0648$ [1].
>
> [1] Michael Johanson, Neil Burch, Richard Anthony Valenzano, Michael Bowling. Evaluating state-space abstractions in extensive-form games. AAMAS 2013: 271-278

---

> > ### Author Response · Authors · 2024-11-21
> > **Rebuttal about questions**
> >
> > Question 1: What is the training time for DeepStack’s replication? Table 4 in the appendix shows that DeepStack takes 569 days, while EVPA takes 74 days. Does this 74 days include the training time for DeepStack’s replication, given that EVPA’s expected value estimation relies on the networks in DeepStack’s replication?
> >
> > **Response Q1**: The $74$ days refer to the combined training time for DeepStack’s replication and EVPA. DeepStack's replication training takes $70$ days, with an additional $4$ days for EVPA's expected value network training ($1$ day for sampling, $3$ days for network training). We have made this clear in Section 4.1 in the revision.
> >
> >     Question 2: What is the element of a subgame S? Line 227 states that $S$ is a set of histories. Line 285 states that an information set I_p is in S. Line 368 states that a public state s(I_p′) is in S.
> >
> > **Response Q2**: A subgame $S$ consists of public states $s$. A public state $s$ corresponds to a number of information sets $I_p$, and an information set $I_p$ corresponds to a number of history states $h$. A history state $h$ corresponds to a public state $s(h)$, and an information set $I_p$ corresponds to a public state $s(I_p)$. These concepts are described in the last paragraph of Section 3.
> >
> >     Question 3: Can you explain the condition 2 in Lines 387-388? Where is I′? Why is the condition needed?
> >
> > **Response Q3**: We apologize for the clerical error regarding $I$'. We have revised it in the revision. The condition ensures that we correctly abstract information sets that are relevant to the current action, preventing overly coarse abstraction at future stages of the game.
> >
> >     Question 4: Why is the significance function in Lines 421-422 needed? The importance of an information set should increase with the size of the pot, but in the function f(g(s),n), it decreases instead.
> >
> > **Response Q4**: The significance function in HUNL normalizes the importance of information sets to avoid mismatches due to the scale of the pot size. This adjustment ensures that the abstraction reflects the relative importance of different stages, especially when the pot size is large.
> >
> >     Question 5: What are the confidence intervals in Table 1 and Figure 3?
> >
> > **Response Q5**: Confidence intervals for pruning ratio and exploitability are negligible, generally less than 1\% of the absolute values. These metrics are stable across multiple experiments.
> >
> >     Question 6: (Line 545) How EVPA is used for any initial chip count? EVPA is based on DeepStack which relies on specific assumptions about players’ initial chips.
> >
> > **Response Q6**: EVPA is flexible with initial chip counts. Unlike DeepStack, which uses fixed chip assumptions, EVPA randomizes the initial chip count in training, within a range of 50–250 big blinds (see Appendix D).
> >
> >     Question 7: Will the code be released to assist research in the field?
> >
> > **Response Q7**: While the EVPA code for HUNL will not be open-sourced due to potential misuse in poker games, we plan to release the code for other domains where EVPA can be applied without the same risks.
> >
> > --------
> >
> > This response addresses all the concerns raised by the reviewer, and we hope the revisions clarify these points. Thank you once again for your thoughtful feedback!  If our responses fully address your concerns, we wonder if you could kindly consider raising your rating score? We will also be happy to answer any further questions you may have. Thank you very much!

---

> > > ### Author Response · Authors · 2024-11-23
> > >
> > > We have revised the paper according to reviewers' insightful comments and helpful suggestions, with revised parts marked in blue. We wonder whether our responses address your concerns? Should you find the rebuttal satisfying, we wonder if you could kindly consider raising the score rating for our paper? We will also be happy to answer any further questions you may have. Thank you very much.

---

> > > > ### Author Response · Authors · 2024-11-25
> > > >
> > > > Dear Reviewer,
> > > >
> > > > Thanks for your time and effort in reviewing our paper.
> > > >
> > > > We hope our response has adequately addressed your concerns. If you feel that our rebuttal has clarified the issues raised, we kindly ask you to consider adjusting your score accordingly. Should you have any further questions or need additional clarification, we would be more than happy to discuss them with you.
> > > >
> > > > Thanks once again for your valuable feedback.
> > > >
> > > > Best regards,
> > > >
> > > > Authors

---

> > > > > ### Comment · Reviewer_85Xa · 2024-11-26
> > > > >
> > > > > I appreciate the authors for the detailed response. I'm raising my score accordingly.

---

> > > > > > ### Author Response · Authors · 2024-11-26
> > > > > >
> > > > > > Thank you very much for raising the score! We really appreciate your positive feedback.

---

### Official Review · Reviewer_jemo · 2024-11-01

**Soundness:** 2
**Presentation:** 3
**Contribution:** 2
**Rating:** 6
**Confidence:** 4

**Summary:**

The authors introduce EVPA, which is an algorithm for approximating Nash equilibria in two-player zero-sum games based on Counterfactual Regret Minimization. EVPA seeks to improve the runtime efficiency of CFR by pruning sub-optimal actions and abstracting information online. This leads to much faster, high-quality approximations of Nash equilibria compared to a replication of DeepStack in Heads-up no-limit Texas Hold 'em poker.

**Strengths:**

* Pruning and abstraction in the context of scaling CFR is highly relevant to the community. Current methods simply do not scale well beyond some of the traditional applications.
* Significant portions of the game tree are successfully pruned, though it's unclear what that means for correctness.
* The overall approach seems to have gained an order of magnitude speedup over DeepStack in both training and online solving time. This suggests that it could be useful in applications that are more time-sensitive than poker.

**Weaknesses:**

* The claim that the method for information abstraction reduces "the computational overhead of information abstraction from months to less than 1 second" seems false. The method for information abstraction depends on the expected value of an information set according to an approximation of the equilibrium strategy. This expected value presumably takes a large portion of the 74 training days reported in the appendix. Would it be more correct to rephrase this to explain the algorithm finds new abstractions online in seconds, given estimates of information set EV?

* No theoretical guarantees. It seems that by pruning and abstracting information using the methods presented in this paper, the approach loses the convergence guarantees of DeepStack. Do the authors believe there is a promising avenue to obtain some similar bounds? The empirical results certainly show that EVPA computes strong strategies in this domain, but are there situations where pruning or abstraction like this can have catastrophic effects on performance?

* Application to a domain that has limited room for further progress. Though the speedup and performance are impressive (see comment above), the application to HUNL has the potential to have less impact on the community than if it were applied to new, more challenging domains, where super-human performance has not already been achieved. Can the authors share some insight on what other sorts of domain this approach could have a significant impact? It seems a game with a higher branching factor could be a good candidate.

**Questions:**

* Can the authors please comment on the quality of Deepstack's replication? How does it perform against local best response (LBR)?

* In the experiments in Table 2, what happens with longer solving times?

* How do we ensure pruning does not severely impact the quality of the final strategy?

* How do we know the method for information abstraction is more applicable to games with large information sets? Or do we specifically mean only Omaha here?

---

> ### Author Response · Authors · 2024-11-21
> **Rebuttal about weaknesses**
>
> Thank you very much for your time and effort in reviewing our paper! Please find our responses to your comments below. We will be happy to answer any further questions you may have.
>
>     Weakness 1. The claim that the method for information abstraction reduces "the computational overhead of information abstraction from months to less than 1 second" seems false. The method for information abstraction depends on the expected value of an information set according to an approximation of the equilibrium strategy. This expected value presumably takes a large portion of the 74 training days reported in the appendix. Would it be more correct to rephrase this to explain the algorithm finds new abstractions online in seconds, given estimates of information set EV?
>
> **Response W1**: We appreciate the reviewer’s observation. After reconsideration, we agree that the original phrasing might cause confusion. Thus, in our revised manuscript, we clarify that the $74$ days of training time reported in the appendix includes the training time of DeepStack’s replication, while the time for EVPA’s sampling and training is only $4$ days in total. We also rephrase the original claim to emphasize that EVPA finds new abstractions online in seconds, based on estimates of the information set's expected value. This more accurately reflects the practical runtime benefits of the method. This clarification has been added in Section 4.1 in the revision.
>
>
>     Weakness 2. No theoretical guarantees. It seems that by pruning and abstracting information using the methods presented in this paper, the approach loses the convergence guarantees of DeepStack. Do the authors believe there is a promising avenue to obtain some similar bounds? The empirical results certainly show that EVPA computes strong strategies in this domain, but are there situations where pruning or abstraction like this can have catastrophic effects on performance?
>
> **Response W2**: We acknowledge that we did not provide theoretical guarantees in the original version of the paper. In the revision, we have added a theoretical analysis of EVPA pruning in Section 4.2. Our analysis shows that in the worst case, the probability of incorrectly pruning a branch that should not be pruned is upper bounded by $O(\frac{|A|-1}{C_{3M}^M})$, where $M$ is the number of neural networks used for expected value estimation, and $|A|$ is the number of branches (with a maximum of $10$). For instance, with $M=10$, the upper bound is $3.0\times 10^{-7}$, and with $M=20$, the upper bound is $2.2\times 10^{-15}$. Our experiments show that the probability of wrong pruning is much lower than these theoretical bounds.
>
> When there is no incorrect pruning, the strategy EVPA produces is at least comparable to DeepStack's replication strategy. We have also provided empirical results showing that EVPA strategies perform well and exhibit low exploitability. These results suggest that the pruning does not degrade strategy soundness, and, as shown by our experiments, there is no evidence of catastrophic performance loss when pruning is applied appropriately.
>
>     Weakness 3. Application to a domain that has limited room for further progress. Though the speedup and performance are impressive (see comment above), the application to HUNL has the potential to have less impact on the community than if it were applied to new, more challenging domains, where super-human performance has not already been achieved. Can the authors share some insight on what other sorts of domain this approach could have a significant impact? It seems a game with a higher branching factor could be a good candidate.
>
> **Response W3**: We believe that EVPA has substantial potential for both pruning and abstraction, beyond just HUNL. For example, the abstraction techniques in EVPA are particularly useful for games with large information sets. We specifically mention Omaha, where the number of information sets is much larger than in HUNL, and the benefits of EVPA's abstraction are even more pronounced. Moreover, EVPA’s approach could be applied outside of poker in domains with manageable public information sets. We agree with the reviewer that games with a higher branching factor could benefit from EVPA's techniques.

---

> ### Author Response · Authors · 2024-11-21
> **Rebuttal about questions**
>
> Question 1. Can the authors please comment on the quality of Deepstack's replication? How does it perform against local best response (LBR)?
>
> **Response Q1**: DeepStack’s replication has been shown to achieve high-quality performance. Specifically, DeepStack's replication beats Slumbot with a win-rate of $32\pm 65$ mbb/h in a $2$-second solving time limit and has an exploitability of no more than $0.01$ after $1\times 10^9$ touches of the information sets. This demonstrates its robustness. Compared to Local Best Response (LBR), DeepStack’s replication outperforms LBR by at least $200$ mbb/h in terms of win-rate.
>
>     Question 2. In the experiments in Table 2, what happens with longer solving times?
>
> **Response Q2**: We have updated the revision to include results for longer solving times in Section 5. Specifically, we provide additional experiments where solving times are extended to $1\times 10^9$ information set touches. When the solving time was further increased to $2$ seconds, DeepStack's replication managed to beat Slumbot with a win-rate of $33\pm 65$ mbb/h, while EVPA beat Slumbot with a win-rate of $100\pm 68$ mbb/h. Meanwhile, when the number of touched information sets came to $1\times 10^9$, DeepStack's replication was still outperformed by EVPA, with recorded a win-rate of $82\pm 60$ mbb/h. The extended solving times demonstrate that EVPA continues to maintain high-quality performance even with longer solving time.
>
>     Question 3. How do we ensure pruning does not severely impact the quality of the final strategy?
>
> **Response Q3**: As mentioned above, our theoretical analysis (Section 4.2) shows that the probability of incorrect pruning is extremely low, even in the worst-case scenario. In practice, pruning is applied conservatively, and extensive empirical evaluation shows that the quality of the final strategy remains high. We ensure that pruning only eliminates clearly suboptimal branches, and through the pruning process, we maintain a strategy that is at least comparable to DeepStack’s replication strategy. Our results also demonstrate that pruning leads to significant speedups with only minimal performance losses.
>
>     Question 4. How do we know the method for information abstraction is more applicable to games with large information sets? Or do we specifically mean only Omaha here?
>
> **Response Q4**:  EVPA’s abstraction algorithm performs close to the theoretical bounds in the early iterations. For example, EVPA-169 ($169$ buckets) reduces exploitability by $54$\% relative to EVPA-full ($1,081$ buckets) with $1\times 10^6$ information set touches. The theoretical exploitability reduction is $1-\sqrt{\frac{169}{1081}}\approx 60$\%. EVPA-30 achieves about $85$\% reduction in exploitability relative to EVPA-full with $1\times 10^6$ touches, while the theoretical bound is $1-\sqrt{\frac{30}{1081}}\approx 83$\%.
>
> These results show that EVPA abstraction is effective even in smaller information sets. However, in games like Omaha, where the number of information sets is at least $100$ times that of HUNL, the advantages of EVPA's abstraction algorithm become even more pronounced. This suggests that EVPA's abstraction technique is particularly suited for games with large information sets. In the case of Liar’s Bar and other complex multi-agent domains, we believe EVPA’s approach can also be very effective.
>
> -----
>
> We hope that the revisions and additional clarifications address the reviewers' concerns. The theoretical analysis of pruning, updated results for longer solving times, and further explanation of EVPA’s applicability to other domains should provide additional confidence in the soundness and potential impact of our approach.
>
> Thank you once again for your thoughtful feedback!  If our responses fully address your concerns, we wonder if you could kindly consider raising your rating score? We will also be happy to answer any further questions you may have. Thank you very much!

---

> > ### Author Response · Authors · 2024-11-23
> >
> > We have revised the paper according to reviewers' insightful comments and helpful suggestions, with revised parts marked in blue. We wonder whether our responses address your concerns? Should you find the rebuttal satisfying, we wonder if you could kindly consider raising the score rating for our paper? We will also be happy to answer any further questions you may have. Thank you very much.

---

> > > ### Author Response · Authors · 2024-11-25
> > >
> > > Dear Reviewer,
> > >
> > > Thanks for your time and effort in reviewing our paper.
> > >
> > > We hope our response has adequately addressed your concerns. If you feel that our rebuttal has clarified the issues raised, we kindly ask you to consider adjusting your score accordingly. Should you have any further questions or need additional clarification, we would be more than happy to discuss them with you.
> > >
> > > Thanks once again for your valuable feedback.
> > >
> > > Best regards,
> > >
> > > Authors

---

> > ### Comment · Reviewer_jemo · 2024-11-26
> >
> > Thank you for your detailed rebuttal and for responding to my questions and concerns. The additional theoretical results are nice, but I still have concerns regarding claims that the advantages of this method become more pronounced in games with more infosets. It doesn't seem feasible to do so during a rebuttal period, but to raise my score further I would need some supporting empirical evidence.

---

> > > ### Author Response · Authors · 2024-11-26
> > >
> > > Thank you for your endorsement of our paper and rebuttal, we will try to test the performance of EVPA on games with more information sets in the next few days!

---

> > > > ### Author Response · Authors · 2024-12-01
> > > >
> > > > Based on reviewer's concern, we tested solving time to reach approximate Nash equilibrium under a river subgame of **PLO5-Hi-Lo**. As the results are shown in the table below, it can be seen that EVPA's abstraction algorithm has a more significant improvement in finding approximate equilibrium on larger subgame, and is able to achieve a speedup of more than $1,000\times$.
> > > >
> > > > |Exploitability|DCFR|EVPA-full|EVPA-1000|
> > > > |---|---|---|---|
> > > > |$<0.01$|$7,131$s|$662$s|$1$s|
> > > > |$<0.001$|$63,372$s|$5,566$s|$4$s|
> > > >
> > > > PLO5-Hi-Lo Rule: Each player is dealt $5$ private hands and checks to each other until the river stage, then the players have the option to check, fold, or bet a pot. There are $5$ blinds in the pot during the river stage and both players have $200$ blinds left.  At showdown the pot is divided into two equal parts, one of which is assigned according to PLO5 rules, and the other is awarded to the player holding the smaller unpaired hand of no more than $8$. Each player has $1,533,939$ different information sets in the river stage.
> > > >
> > > > Thank you once again for your valuable feedback! We hope the empirical result in PLO5-Hi-Lo could address your concern.

---

### Official Review · Reviewer_Ce7c · 2024-11-02

**Soundness:** 2
**Presentation:** 2
**Contribution:** 2
**Rating:** 3
**Confidence:** 4

**Summary:**

***Summary of paper** This paper proposes three techniques to improve CFR: (i) expected value estimation of information sets (ii)expected value-based pruning and (iii)information abstraction for subgames. The authors also gave experimental evaluation against an implementation of DeepStack and SlumBot.

**Strengths:**

The presentation is clear.

**Weaknesses:**

Some technical points are not sound.

**Questions:**

**Detailed questions and comments.** For the three techniques this paper proposes, I have several concerns and questions.

1. Expected value estimation of information sets: How does it compare with the one used in DeepStack? How do you compute Nash of subgame? Is it exact?  One problem of DeepStack is it generates subgames by random sampling, which may not cover the most essential ones. A more advanced one is used in Student of Games which estimates values by considering those encounter during CFR iteration recursively. Have you tried these alternative.


2. Min-max pruning. I don't think min-max pruning is correct in imperfect information games unless you transform the game into public belief tree representation, which you should include public belief states into the value function. However as far as I see the value function you are using are information-set value function, which is incorrect to use min-max pruning because the opponent can change the reach probabilities by changing its action prior to the current information set. However, it you were use min-max pruning for public-belief tree, it will be much more computationally expensive.

3. Abstraction: How often is the abstraction criterion met? How does it compare with previous abstraction techniques such as nested subgame solving in [Brown & Sandholm '17]

---

> ### Author Response · Authors · 2024-11-21
> **Rebuttal of Expected value estimation of information sets**
>
> Thank you very much for your time and effort in reviewing our paper! Please find our responses to your comments below. We will be happy to answer any further questions you may have.
>
>     Question 1: Expected value estimation of information sets: How does it compare with the one used in DeepStack? How do you compute Nash of subgame? Is it exact? One problem of DeepStack is it generates subgames by random sampling, which may not cover the most essential ones. A more advanced one is used in Student of Games which estimates values by considering those encounter during CFR iteration recursively. Have you tried these alternative.
>
> **Response 1**: EVPA's expected value estimation differs significantly from DeepStack's in the following aspects.
>
> **(Public state)** DeepStack uses a public belief state as input to its neural network, which includes both players' ranges. The output is an expected value based on these ranges. In contrast, EVPA uses a public state that includes only observable information such as action history, chip counts, public cards, and each player's private hand. It does not include the players' belief states or ranges. The output of EVPA is the expected value corresponding to the DeepStack replication for that public state, which we view as an approximation of the Nash equilibrium.
>
> **(Sampling)** While DeepStack generates subgames via random sampling, which may not always capture all important branches, we ensure that our sampling method in DeepStack's replication (Appendix D) and EVPA sampling (Section 4.1) addresses this concern.
>
> For DeepStack's replication sampling, we use a method inspired by ReBeL and Supremus, starting with random sampling and then switching to a more efficient ReBeL-like strategy for generating data during training. This approach improves coverage, especially for both essential and rare parts of the game tree.
>
> For EVPA sampling, Algorithm 1 in Section 4.1 specifies our sampling procedure, which ensures broad coverage by randomly selecting branches in the game tree, incoporating initial states, action abstractions, and sampling from leaf nodes. We compute the approximate Nash equilibrium using DeepStack's replication combined with CFR-based techniques.
>
> **(Regarding Student of Games)** Regarding Student of Games, we have not directly tested this method. However, our method incorporates similar ideas from ReBeL [1], which has shown strong performance in poker. We believe EVPA's sampling method provides similar or better coverage, as evidenced by the results from our experimental evaluation.
>
> [1] Noam Brown, Anton Bakhtin, Adam Lerer, Qucheng Gong. Combining Deep Reinforcement Learning and Search for Imperfect-Information Games. NeurIPS 2020

---

> ### Author Response · Authors · 2024-11-21
> **Rebuttal of Min-max pruning**
>
> Question 2: Min-max pruning. I don't think min-max pruning is correct in imperfect information games unless you transform the game into public belief tree representation, which you should include public belief states into the value function. However as far as I see the value function you are using are information-set value function, which is incorrect to use min-max pruning because the opponent can change the reach probabilities by changing its action prior to the current information set. However, it you were use min-max pruning for public-belief tree, it will be much more computationally expensive.
>
> **Response 2**: We appreciate your concern and would like to clarify that our approach to min-max pruning in EVPA is sound and carefully designed for IIEFGs. You are correct that min-max pruning is challenging in these games, especially during CFR iterations. However, EVPA is designed specifically to handle this issue, as **it performs permanent and correct pruning before CFR starts, not during the iteration process.**
>
> In Section 4.2, we explain the motivation for pruning in EVPA. For example, in HUNL, we would never consider folding a pair of aces in the pre-flop stage, and there is no need to compute these branches during CFR iterations. EVPA performs pruning before CFR solving, which helps reduce computational complexity without introducing errors due to changing reach probabilities during CFR iterations.
>
> Regarding your point on public-belief trees, we agree that incorporating public belief states into the value function would be computationally expensive. However, EVPA's approach does not require this transformation. Instead, EVPA uses neural networks trained on public information to estimate expected values for information sets before the solving process begins. These estimates are used for pruning and abstraction, and the resulting pruning decisions are shown to have minimal impact on the final strategy.
>
> In the revision, our analysis (detailed in Section 4.2) shows that the theoretical probability of incorrect pruning is very low. Specifically, in the worst case, this probability is upper bounded by $O(\frac{|A|-1}{C_{3M}^M})$, where $M$ is the number of neural network used for expected value estimation, and $|A|\leq 10$ is the number of branches. For example, when $M=10$, the upper bound is $3.0\times 10^{-7}$, and when $M=20$, the bound is $2.2\times 10^{-15}$. Our experimental results confirm that the probability of incorrect pruning is even lower than these bounds.
>
> In summary, min-max pruning in EVPA is applied before CFR solving and is theoretically sound. Our experiments also validate that this method significantly improves computational efficiency without negatively affecting strategy quality.

---

> ### Author Response · Authors · 2024-11-21
> **Rebuttal of Abstraction**
>
> Question 3. Abstraction: How often is the abstraction criterion met? How does it compare with previous abstraction techniques such as nested subgame solving in [Brown \& Sandholm '17]
>
> **Response 3**: We evaluated abstraction accuracy in terms of Euclidean distance within buckets. After normalizing the features, we observed the following average distances to the bucket center was $0.015$ and $0.083$ for EVPA-169 and EVPA-30, respectively. These results indicate that the abstraction criterion is met quite well.
>
> Regarding comparison with previous techniques, EVPA is based on the nested subgame solving technique, as you correctly pointed out. In Section 5, we use nested subgame solving as a baseline and demonstrate that EVPA’s abstraction approach performs very close to the theoretical bounds in early iterations. For example: EVPA-169 ($169$ buckets) reduces exploitability by $54$\% compared to EVPA-full ($1,081$ buckets) with $1\times 10^6$ information set touches. The theoretical exploitability reduction is only $1-\sqrt{\frac{169}{1081}}\approx 60$\%. EVPA-30 ($30$ buckets) achieves about $85$\% reduction in exploitability relative to EVPA-full with $1\times 10^6$ touches, while the theoretical bound is $1-\sqrt{\frac{30}{1081}}\approx 83$\%. These results demonstrate that EVPA's abstraction is highly efficient and nearly optimal in terms of exploitability. We have also compared EVPA’s pruning and abstraction methods with previous methods in Appendices G and H of the revision.
>
> -----
>
> We hope that our revision and responses address the concerns raised by the reviewer, particularly regarding the correctness of min-max pruning in imperfect information games, the robustness of expected value estimation, and the effectiveness of our abstraction techniques.
>
> We also would like to express our gratitude to the reviewer again for the effort in reviewing our
> paper. If our responses fully address your concerns, we wonder if you could kindly consider raising your rating score? We will also be happy to answer any further questions you may have. Thank you very much!

---

> > ### Author Response · Authors · 2024-11-23
> >
> > We have revised the paper according to reviewers' insightful comments and helpful suggestions, with revised parts marked in blue. We wonder whether our responses address your concerns? Should you find the rebuttal satisfying, we wonder if you could kindly consider raising the score rating for our paper? We will also be happy to answer any further questions you may have. Thank you very much.

---

> > > ### Author Response · Authors · 2024-11-25
> > >
> > > Dear Reviewer,
> > >
> > > Thanks for your time and effort in reviewing our paper.
> > >
> > > We hope our response has adequately addressed your concerns. If you feel that our rebuttal has clarified the issues raised, we kindly ask you to consider adjusting your score accordingly. Should you have any further questions or need additional clarification, we would be more than happy to discuss them with you.
> > >
> > > Thanks once again for your valuable feedback.
> > >
> > > Best regards,
> > >
> > > Authors

---

> > > > ### Author Response · Authors · 2024-11-27
> > > >
> > > > We hope our responses fully address your concerns. If so, we wonder if you could kindly consider raising your score rating? Meanwhile, we would be more than happy to answer any further questions. Thanks once again for your valuable feedback.

---

### Official Review · Reviewer_cDdZ · 2024-11-03

**Soundness:** 2
**Presentation:** 2
**Contribution:** 3
**Rating:** 8
**Confidence:** 5

**Summary:**

This paper introduces a method called EVPA, for online pruning and abstraction of imperfect-information games (IIGs) based on an evaluation of information sets, which is trained offline, in a preparation of an online play.
The evaluation is done by a collection of neural networks, to which the same information set is used as input.
The collected values serve as an estimate of minimum, maximum, and average values that can encountered for that information set in online play, when solving a subgame.
By employing a minimax-like pruning, a subgame can be reduced in size, in advance of solving that subgame, which reduces the time needed for the online solving.
This contrasts with algorithms like CFR, for which pruning has been done in prior work. CFR prunes the tree only dynamically, based on the current state of the algorithm, and still requires full subgame at the first iteration.
Additionally, if the estimates of values for structurally-compatible information sets are similar, they can be merged together. This is similar to bucketing done in prior work.
Finally, EVPA is used for large variants of Poker and its performance is measured empirically against other prior strong algorithms. EVPA can find strong strategies while requiring substantially smaller time to do so.

**Strengths:**

- A simple idea that should be easy to implement and can be generally incorporated into many algorithms, as it mainly concerns a general structural aspect of IIGs.
- Strong performance on poker games.

**Weaknesses:**

- The paper offers only empirical results. It makes very little evaluations on choices needed for the offline preparation, and only considers the online parameters like time budget. However, a practitioner has to decide a number of choices for the offline preparation: the number of neural networks, the size of training set, a choice of the delta parameter. It is not clear how the approach is robust to these. While I understand this is computationally expensive, it can be done for some (limited) choices, while not making the budget too high. Looking at appendix, DeepStack (which has been replicated in the paper) training time is 569 days, while EVPA is 74. Spending up to 7x budget on more evaluations thus seems reasonable.

- Establishing theoretical bounds based on the choice of delta and EV estimate errors should not be difficult, but is not present in the paper. However, the approach does seem sound.

- The text is a bit repetitive. Instead, I would appreciate more involved discussion of prior abstraction literature and how this work differs.

**Questions:**

- L300 "each neural network receives a feature describing the information set" --- What exactly comprises the feature? The value of an infoset is not uniquely defined without the context of ranges for a public state. Is this information included?
- Can you please add results where the time limit is increased, up to 2 and 20 seconds, and against DeepStack, for a time budget?
- For the heads-up results, how many matches have been played? Have you considered using AIVAT to reduce variance?

---

> ### Author Response · Authors · 2024-11-21
> **Rebuttal about weaknesses**
>
> Thank you very much for your time and effort in reviewing our paper! Please find our responses to your comments below. We will be happy to answer any further questions you may have.
>
>
>         Weakness 2. Establishing theoretical bounds based on the choice of delta and EV estimate errors should not be difficult, but is not present in the paper. However, the approach does seem sound.
>
> Response W2: We appreciate the reviewer's suggestion. Following your comment, we have added a theoretical analysis of EPVA pruning in the revision (see Section 4.2 in the revision). Specifically, we provide an upper bound on the probability of incorrectly pruning a branch that should not be pruned. This probability is given by $O(\frac{|A|-1}{C_{3M}^M})$, where $M$ is the number of neural networks used for expected value estimation and $|A|$ is the number of branches (which has a maximum value of $10$). For $M=10$, this bound is  $3.0\times 10^{-7}$, and for $M=20$, the bound is $2.2\times 10^{-15}$. In practice, the probability of incorrect pruning in our experiments is even lower than these theoretical estimates.
>
>     Weakness 1. The paper offers only empirical results. It makes very little evaluations on choices needed for the offline preparation, and only considers the online parameters like time budget. However, a practitioner has to decide a number of choices for the offline preparation: the number of neural networks, the size of training set, a choice of the delta parameter. It is not clear how the approach is robust to these. While I understand this is computationally expensive, it can be done for some (limited) choices, while not making the budget too high. Looking at appendix, DeepStack (which has been replicated in the paper) training time is 569 days, while EVPA is 74. Spending up to 7x budget on more evaluations thus seems reasonable.
>
> Response W1: We agree with the reviewer that the choice of offline preparation parameters (such as the number of neural networks and the delta parameter) is important.  Our theoretical analysis and experiments show that setting $M=10$ and $\delta=0.01$ offers a good balance between pruning effectiveness and correctness. The table below summarizes the results of additional experiments on hyperparameter selection, showing the pruning rate and correctness (``Yes'' if the exploitability could smaller than $0.001$ in all of $1,000$ river stage subgame samples) for different values of $M$ and $\delta$.
>
> | | $M=1$ | $M=5$ | $M=10$ |
> |:------:|:------:|:-------:|:------:|
> |$\delta=0$| $99$%,No | $89$%,No | $86$%,Yes |
> |$\delta=0.01$| $96$%,No | $85$%,Yes | **80%,Yes** |
> |$\delta=0.05$| $80$%,No | $74$%,Yes | $66$%,Yes |
>
> These results show that with $M=10$ and $\delta=0.01$, the strategy remains sound with good pruning effectiveness. Additionally, we clarify in Section 4.1 of the revised manuscript that the additional training time for EVPA is only $4$ days.
>
>     Weakness 3. The text is a bit repetitive. Instead, I would appreciate more involved discussion of prior abstraction literature and how this work differs.
>
> Response W3: We appreciate the feedback. In response, we have added a more detailed comparison to prior abstraction methods in Appendix H. Specifically, EVPA offers two key advantages over previous work: (i)  Online Abstraction: The abstraction process takes less than 1 second, allowing it to be performed dynamically during subgame solving. (ii) Efficient Performance: The abstraction process produces strong results with fewer buckets, demonstrating a significant reduction in exploitability.
>
> For example, EVPA-169 (with $169$ buckets) achieves a $54$\% reduction in exploitability relative to EVPA-full ($1,081$ buckets) with $1\times 10^6$ information set touches. The theoretical exploitability reduction is only $1-\sqrt{\frac{169}{1081}}\approx 60$\%. EVPA-30 (with $30$ buckets) reduces exploitability by approximately $85$\% relative to EVPA-full, with a theoretical reduction of $1-\sqrt{\frac{30}{1081}}\approx 83$\%. These results demonstrate that EVPA's abstraction is highly efficient and nearly optimal in terms of exploitability.

---

> > ### Comment · Reviewer_cDdZ · 2024-11-28
> >
> > Thank you for the update! I do not follow what $C_{3M}^M$ refers to - it seems it's not defined anywhere.
> > I like the follow up and after this is bound is resolved I will update my rating.

---

> > > ### Author Response · Authors · 2024-11-28
> > >
> > > Thanks to reviewer's feedback, our analysis on the pruning correctness bound is in Section4.2, Line 366-386 in the revision. Pruning error occurs only if the $M$ estimate values of the optimal branch are smaller than the $2M$ estimates of both its father and another optimal sibling. The probability of selecting the exactly $M$ minimum values out of $3M$ random values is $\frac{1}{C_{3M}^M}$. We hope that the analyses will address the reviewer's concern, thanks again.

---

> > > > ### Comment · Reviewer_cDdZ · 2024-11-29
> > > >
> > > > What is the number $C_{3M}^M$, resp. $C_{3M}$ ? Do we know it's value?

---

> ### Author Response · Authors · 2024-11-21
> **Rebuttal about questions**
>
> Question 1. L300 "each neural network receives a feature describing the information set" --- What exactly comprises the feature? The value of an infoset is not uniquely defined without the context of ranges for a public state. Is this information included?
>
> Response Q1: The features used by the expected value estimation neural networks consist only of public information, including previous actions, positions, chip counts, public card information, and the player's hand. We explicitly state this in Section 4.1 of the revision. As the reviewer correctly points out, the value of an information set cannot be uniquely defined without considering belief state ranges, which change during CFR iterations. However, the expected value estimation neural networks in EVPA are used only for pruning and abstraction **before** CFR solving. The absence of belief state information does not affect the soundness of the final strategy because the pruning is performed based on DeepStack's replication strategy, and the probability of incorrect pruning is extremely low, as discussed earlier.
>
>
>
>     Question 2. Can you please add results where the time limit is increased, up to 2 and 20 seconds, and against DeepStack, for a time budget?
>
> Response Q2: In response to your comment, we provide new results for a $2$-second time limit against Slumbot and for a large number of information set touches ($1\times 10^9$) against DeepStack's replication in the revision. When the solving time was further increased to $2$ seconds, DeepStack's replication managed to beat Slumbot with a win-rate of $33\pm 65$ mbb/h, while EVPA beat Slumbot with a win-rate of $100\pm 68$ mbb/h. Meanwhile, when the number of touched information sets came to $1\times 10^9$, DeepStack's replication was still outperformed by EVPA, with recorded a win-rate of $82\pm 60$ mbb/h.
>
> On the other hand, due to time constraints, the $20$-second time limit prevents meaningful results within the rebuttal period, as only about $1,000$ hand evaluations can be performed per day. Based on our estimates, we expect that increasing the time limit to $20$ seconds would result in a win-rate of approximately $10$-$20$ mbb/h compared to the baseline.
>
>     Question 3. For the heads-up results, how many matches have been played? Have you considered using AIVAT to reduce variance?
>
> Response Q3: We indeed used AIVAT to reduce variance in the heads-up evaluations, as mentioned in Section 5 Paragraph 1. The number of hands played in the heads-up evaluation ranged from $20,000$ to $200,000$ hands.
>
> --------------
>
> We hope that the revisions and additional clarifications address the reviewers' concerns. The theoretical analysis of pruning, updated results for longer solving times, and further discussion of other abstraction methods should provide additional confidence in the soundness and potential impact of our approach.
>
> We also would like to express our gratitude to the reviewer again for the effort in reviewing our paper. If our responses fully address your concerns, we wonder if you could kindly consider raising your rating score? We will also be happy to answer any further questions you may have. Thank you very much!

---

> > ### Author Response · Authors · 2024-11-23
> >
> > We have revised the paper according to reviewers' insightful comments and helpful suggestions, with revised parts marked in blue. We wonder whether our responses address your concerns? Should you find the rebuttal satisfying, we wonder if you could kindly consider raising the score rating for our paper? We will also be happy to answer any further questions you may have. Thank you very much.

---

> > > ### Author Response · Authors · 2024-11-25
> > >
> > > Dear Reviewer,
> > >
> > > Thanks for your time and effort in reviewing our paper.
> > >
> > > We hope our response has adequately addressed your concerns. If you feel that our rebuttal has clarified the issues raised, we kindly ask you to consider adjusting your score accordingly. Should you have any further questions or need additional clarification, we would be more than happy to discuss them with you.
> > >
> > > Thanks once again for your valuable feedback.
> > >
> > > Best regards,
> > >
> > > Authors

---

> > > > ### Author Response · Authors · 2024-11-27
> > > >
> > > > We hope our responses fully address your concerns. If so, we wonder if you could kindly consider raising your score rating? Meanwhile, we would be more than happy to answer any further questions. Thanks once again for your valuable feedback.

---

> ### Author Response · Authors · 2024-11-30
>
> This is the combinatorial number, i.e. $\frac{1}{C_{3M}^M}=\frac{M!(2M)!}{(3M)!}$. When $M=10$, this value is around $3*10^{-8}$. We hope that this will address the reviewer's concern and adjust the score of accordingly. Thanks once again for your feedback.

---

> > ### Comment · Reviewer_cDdZ · 2024-12-01
> >
> > Thank you!
> >
> > One more thing: the explotability computation is undefined for the pruned subgame game strategy. I assume you impute uniform strategy for the missing information sets?

---

> > > ### Author Response · Authors · 2024-12-01
> > >
> > > Thanks to your valuable feedback!
> > >
> > > When calculating exploitability, we will calculate exploitability on the subgame without pruning. This is done by setting the current player to be $p$. If $\sigma(I_p,a)=0$, we do not need to compute the exploitability of the $I_p\cdot a$ branch, since this branch contributes $0$ to the exploitability. If $\sigma(I_{-p},a)=0$, we allow the opponent to choose this branch to exploit our strategy. When computing exploitability, we still compute the strategies of both players of the $I_{-p}\cdot a$ subgame. Although these strategies are not computed when EVPA is actually run, we use safe and nested subgame solving techniques that allow us to still compute the strategy of the corresponding subgame in real time when this subgame is touched, and the exploitability does not deteriorate when this strategy is merged into the whole game.
> > >
> > > We hope that the rebuttal will address the reviewer's concern, thanks again.

---

> > > > ### Comment · Reviewer_cDdZ · 2024-12-01
> > > >
> > > > Thank you for the clarification. As all my concerns have been addressed and I believe there will be interesting follow-ups on this work that extend it to much larger games than Poker, I am raising my score.

---

> > > > > ### Author Response · Authors · 2024-12-01
> > > > >
> > > > > Thank you very much for your valuable suggestions and positive feedback! We also believe that EVPA can be extended to games larger than poker and are working towards it!

---

### Meta-Review · Area_Chair_q3JL · 2024-12-21

**Metareview:**

The paper introduces EVPA, a method designed to improve computational efficiency in solving large Imperfect Information Extensive-Form Games (IIEFGs). EVPA combines three key techniques: expected value estimation of information sets, expected value-based pruning, and online abstraction for subgames. Experimental results on Heads-Up No-Limit Texas Hold'em (HUNL) demonstrate that EVPA achieves superior performance over DeepStack’s replication and Slumbot, with significant speedups.

Reviewers cDdZ and jemo highlighted that EVPA offers a significant reduction in computation time and achieves strong strategies, with cDdZ stating, “there will be interesting follow-ups on this work that extend it to much larger games than Poker.” Reviewer 85Xa also emphasized the practical relevance of the method, noting its ability to prune approximately 75% of the game tree while maintaining strong performance.

Reviewer Ce7c raised concerns about the technical soundness of the method, particularly regarding the correctness of min-max pruning in IIEFGs; the authors provided detailed rebuttals; Reviewer Ce7c did not engage further in the discussion or acknowledge the authors' explanations at a technical level. I found the author's response convincing.

Given the theoretical soundness, strong empirical results, and support from experts on IIEFG solving, I recommend acceptance.

**Additional Comments On Reviewer Discussion:**

Summarized above

---

### Decision · Program_Chairs · 2025-01-22

Accept (Poster)